# The influence of viscous slab rheology on numerical models of subduction

Natalie Hummel[1,2], Susanne Buiter[1,3], and Zoltán Erdős[3]

[1]Tectonics and Geodynamics, RWTH Aachen University, Aachen, Germany, 52062
[2]Woods Hole Oceanographic Institution, Falmouth, MA, USA, 02543
[3]Helmholtz Centre Potsdam GFZ German Research Centre for Geosciences, Potsdam, Germany, 14473

**Correspondence:** Natalie Hummel (natalie.hummel@whoi.edu)

**Abstract.**

Numerical models of subduction commonly use diffusion and dislocation creep laws from laboratory deformation experiments to determine the rheology of the lithosphere. The specific implementation of these laws varies from study to study, and the impacts of this variation on model behavior have not been thoroughly explored. We run simplified 2D numerical models of free subduction in SULEC, with viscoplastic slabs following: 1) a diffusion creep law, 2) a dislocation creep law, and 3) both simultaneously, as well as several variations on model 3 with reduced resistance to bending. We compare the results of these models to a model with a constant-viscosity slab to determine the impact of the implementation of different lithospheric flow laws on subduction dynamics. In creep-governed models, higher subduction velocity causes a longer effective slab length, increasing slab pull and asthenospheric drag, which, in turn, affect subduction velocity. Numerical and analogue models implementing constant viscosity slabs lack this feedback, but still capture morphological patterns observed in more complex models. Dislocation creep is the primary deformation mechanism throughout the subducting lithosphere in our models. However, both diffusion and dislocation creep predict very high viscosities in the cold core of the slab. At the trench, the effective viscosity is lowered by plastic failure, rendering effective slab thickness the primary control on bending resistance and subduction velocity. However, at depth, plastic failure is not active, and the viscosity cap is reached in significant portions of the slab. The resulting high slab stiffness causes the subducting plate to curl under itself at the mantle transition zone, affecting patterns in subduction velocity, slab dip, and trench migration over time. Peierls creep and localized grain size reduction likely limit the stress and viscosity in the cores of real slabs. Numerical models implementing only power-law creep and neglecting Peierls creep are likely to overestimate the stiffness of subducting lithosphere, which may impact model results in a variety of respects.

## 1 Introduction

Several mechanisms work in parallel to accommodate deformation in Earth's oceanic lithosphere. On short time scales (hundreds of thousands of years or less), strain takes place mostly by elastic deformation. On longer time scales, strain is dominated by non-recoverable deformation via discrete breaks (brittle failure) and via several aseismic, microscopic mechanisms, including diffusion creep, dislocation creep, and Peierls creep. Diffusion creep is the translation of individual atoms or vacancies through mineral grains (Nabarro-Herring creep) or along grain boundaries (Cobble creep). Dislocation creep is migration of

linear imperfections through a crystal lattice. Diffusion and dislocation creep produce strain rates proportional to the applied stress raised to an exponent of approximately 3.5 or 1, respectively (Hirth and Kohlstedt, 2003; Karato and Wu, 1993). Peierls creep also takes place by the migration of dislocations, but acts as a form of low-temperature plasticity due to its high stress dependence and weak temperature dependence, rather than following a power-law relationship with stress (Guyot and Dorn, 1967).

The relative importance of each deformation mechanism varies over time and space with temperature, pressure, grain size, stress, and water content. Brittle failure is typically dominant in the upper $\sim 30$ km (Turcotte and Schubert, 2014) of the oceanic lithosphere, where low normal stress allows fractures to form and low temperatures prevent creep from taking place. Dislocation creep is thought to dominate deformation in the upper mantle below the brittle ductile transition (Karato et al., 2001; van Hunen et al., 2005; Garel et al., 2014). However, diffusion creep may play a role in other, deeper areas that are cold

or have small grain sizes (Karato and Wu, 1993; van Hunen et al., 2005). Peierls creep is likely active only in areas of very high stress (>500 MPa), such as the cold cores of subducting lithosphere (Kameyama et al., 1999), where other creep mechanisms predict very strong behavior.

Numerical modelers have approximated the rheological properties of subducting lithosphere in a variety of ways. The simplest approach is to implement constant-viscosity slabs 2 or 3 orders of magnitude more viscous than the surrounding astheno-

sphere (Capitanio et al., 2007; Heuret et al., 2007; Kaus and Becker, 2008; Quinquis et al., 2011; Schmeling et al., 2008). This elegantly allows first-order behaviour of subducted slabs to be investigated. More commonly, the rheology of a subducting slab is set to mimic the extrapolated behavior predicted by laboratory deformation experiments on single minerals, monomineralic aggregates, or mantle rock types. These laboratory experiments (Chopra and Paterson, 1981; Kirby, 1983; Wilks and Carter, 1990; Karato and Wu, 1993; Hirth and Kohlstedt, 2003) quantify the stress-strain relationships of individual creep mechanisms

and their dependence on relevant factors such as temperature, pressure, grain size and water content. Numerical modelers have taken various approaches to implementing the resulting flow laws. For example, Tagawa et al. (2007) model the lithosphere using Newtonian, temperature-dependant (diffusion) creep based on data from Karato and Wu (1993), with the pre-exponential factor adjusted to produce an average viscosity of $5 * 10^{20}$ Pas in the upper mantle, and Erdős et al. (2021) use a wet olivine dislocation creep law from Karato and Wu (1993). Quinquis and Buiter (2014) take a slightly more complex approach fol-

lowing van den Berg et al. (1993), using laws determined by Hirth and Kohlstedt (2003) on wet olivine aggregates to model diffusion and dislocation creep simultaneously, such that the strain rates predicted by each mechanism are added to achieve the total strain rate. Arcay (2012) models oceanic lithosphere with an even more complex rheological structure, including regions of dry granulite (Wilks and Carter, 1990), wet dunite (Chopra and Paterson, 1981), dry diabase (Kirby, 1983), and wet olivine (Karato et al., 2001; Hirth and Kohlstedt, 2003).

Using values from laboratory flow laws, which are extrapolated from laboratory time and spatial scales to subduction scales, generally leads to high viscosity values in the interior of cold subducted slabs. Independent of the question of whether such high viscosity values occur in nature, many modeling softwares cannot effectively handle the resulting large variations in viscosity. For this reason, models generally use a maximum stress or viscosity cap. The latter varies from $10^{23}$ Pas (Billen et al., 2003; Behr et al., 2022), $10^{24}$ Pas (Torii and Yoshioka, 2007; Quinquis and Buiter, 2014; Biemiller et al., 2019), $10^{25}$ Pas (Gerya

et al., 2021; Tagawa et al., 2007), to $10^{26}$ Pas (Tetreault and Buiter, 2012; Khabbaz Ghazian and Buiter, 2013). Alternatively, modelers may impose a maximum stress on the order of 500 MPa, which roughly approximates the effect of Peierls creep (Ĉížková et al., 2002; Behr et al., 2022), or may implement Peierls creep more precisely (Garel et al., 2014). Though both stress and viscosity caps limit the strength of the lithosphere, they do not generally produce the same slab behavior (Billen, 2008).

The general mechanics of subduction have been investigated through numerous analogue and numerical experiments. Funiciello et al. (2008) document 4 stages in analogue models of free subduction: (1) subduction initiates and (2) the slab tip sinks through the upper mantle with increasing velocity until (3) subduction slows temporarily as the slab interacts with the bottom of the tank, and (4) eventually reaches a steady-state, with the end of the slab lying flat on the bottom of the tank and trench retreat proceeding at a constant velocity.

The details of this subduction process–the evolution of slab dip, subduction velocity, and trench motion–are affected by the rheologic structure of the subducting plate. Several studies (Billen and Hirth, 2007; Capitanio et al., 2007, 2009; Ĉížková et al., 2002; Arcay, 2012; DiGiuseppe et al., 2008; Kaus and Becker, 2008; Garel et al., 2014; Ribe, 2010) have explored the relationship between subduction behavior and the bending resistance of the subducting lithosphere, which is proportional to the cube of slab thickness and the slab's viscosity contrast with the surrounding mantle. Increasing the slab-mantle viscosity contrast has been shown to decrease slab dip and increase subduction velocity in the steady-state stage of subduction (Capitanio et al., 2007). DiGiuseppe et al. (2008) show that slabs with higher bending resistance are more prone to trench advance. The numerical models of Kaus and Becker (2008), which simulate constant-viscosity lithosphere and no overriding plate, illustrate that subducting plates of low viscosity ($10^{21}$ - $10^{22}$ Pas) bend at the trench and unbend to subduct forward at a steep angle into the upper mantle, as observed by Funiciello et al. (2008), whereas high viscosity plates ($10^{23}$ - $10^{24}$ Pas) are too stiff to unbend, instead keeping a side-ways U-shape or curl. In such cases, slab pull is too low and bending resistance too high to achieve unbending (Goes et al., 2017; Stegman et al., 2010). Ĉížková et al. (2002) show that changing grain size or the stress cap (and by extension the bending resistance) in the cold core of a slab affects its interaction with the mantle transition zone; stiff plates penetrate into the lower mantle, and weak plates bend forward, consistent with the analysis conducted by Ribe (2010) and the results of numerical models presented by Garel et al. (2014).

Studies have also explored how the heterogeneous structure of subducting lithosphere produces behavior that is not observed with constant-viscosity approximations. For instance, Capitanio et al. (2009) demonstrate that plates require a strong, thin (less than the typical thickness of an oceanic plate) core to bend readily at the trench, yet maintain sufficient resistance to stretching to transmit stress from the slab to the surface. Garel et al. (2014) point out feedbacks between subduction velocity, slab strength, and slab pull that complicate subduction in models with dynamic slabs, and Androvičová et al. (2013) illustrate how a low-viscosity crust is necessary to decouple the subducting and overriding plates and achieve realistic subduction. However, there are further subtleties of slab structure to explore, such as the ways in which creep laws control slab bending resistance at the trench and at depth in dynamic models.

Garel et al. (2014) map out the importance of various deformation mechanisms spatially across a subduction zone, with Peierls creep and maximum viscosity limits active in large portions of their modeled slabs. These mechanisms impact bending

resistance and, by extension, plate behavior. Despite this, it is common for numerical models to implement diffusion and dislocation creep without stress limiting mechanisms like Peierls creep. The cold cores of these slabs typically reach the viscosity maximum of the model, which can vary by several orders of magnitude between models.

It is clearly important to understand how the choice of flow law in a numerical model affects slab rheology and, by extension, subduction dynamics. In this study, we compare the behavior of simplified numerical models of subduction with variable slab rheologies (diffusion-creep only, dislocation-only, and diffusion and dislocation together). We also explore plate weakening through reduced grain size and a lowered viscosity cap. We analyze how subduction dynamics predicted by each approach compare to the behavior of constant-viscosity models and to real subduction zones, breaking down the impact of slab rheology and bending resistance near the surface and at depth on plate behavior as subduction progresses. We hope that these experiments raise awareness of the limitations of using extrapolated flow laws in numerical models of subduction and initiate a discussion on high viscosity values reached in many models.

## 2   Model Set Up

In order to investigate the effects of different flow laws on slab rheology and behavior, we use 2D models of free subduction with variable flow laws active in the slab. The models are highly simplified—self-consistent with linear viscous crust and mantle—in order to focus on the effects of slab rheology. We implement individual flow laws to illustrate the contribution of each law to deformation in the slab and investigate whether increasing rheological complexity in the slab has significant implications for model behavior.

We run experiments in SULEC, an Arbitrary Lagrangian Eulerian finite element code (Buiter and Ellis, 2012) using the PARDISO solver (Schenk and Gartner, 2004). SULEC solves conservation of energy and momentum equations for an incompressible fluid and advects tracers recording material properties through an element mesh of prescribed density. We use Courant time stepping with a Courant number of 0.1 and apply a weak diffusive erosion process with a diffusion coefficient of $10^{-6}$ $m^2 s^{-1}$ (Culling, 1960) to limit surface instabilities.

The initial model geometry and temperature field are identical in all models (Fig. 1). The models are 3080 km wide by 660 km deep and have nodes spaced every 6 km in the horizontal direction, with a finer spacing around the trench (Fig. 1). The vertical node spacing increases from 2 km at the surface to 6 km below 240 km depth. The elements have 4 nodes for velocity and are constant in pressure. The subducting lithosphere is 80 km thick and 1430 km long from trench to trailing end, with an 8 km-thick crust (Fig. 1). The models have a free surface and the sides and bottom of the models are free slip. No material enters or exits the model domain. We do not impose a pushing force on the plate. Instead, subduction is driven by the density contrast between the lithosphere and the asthenosphere. In the initial set-up, a small section of the slab tip dips under the overriding plate at an angle of 30° to a depth of 183 km to facilitate subduction initiation. The subducting plate lacks crust along a 100 km-long section of its trailing end, and subduction stalls when this section reaches the trench, marking the end of the experiment. We leave 100 km of asthenosphere on either side of the lithosphere to allow the plates to slide horizontally.

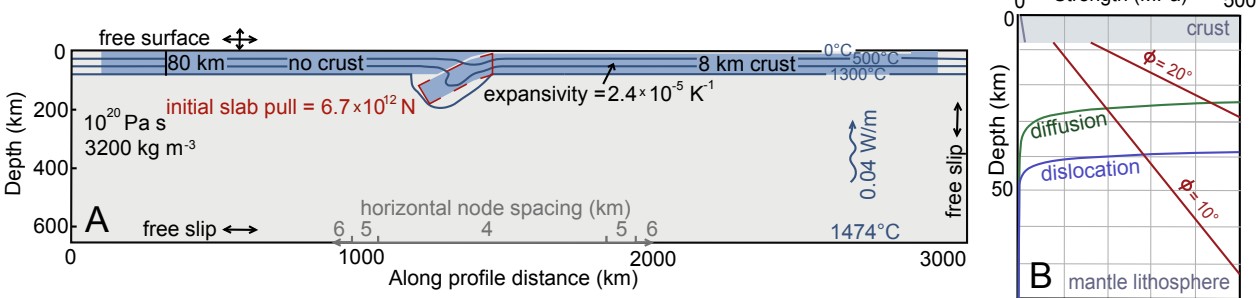

**Figure 1.** The initial set-up for all models. A shows the oceanic lithosphere in blue and the asthenosphere in grey, with thermal parameters in dark blue text and kinematic parameters and material properties in black text. The red box illustrates the area over which the slab density contrast compared to a background asthenospheric density profile is integrated to obtain the slab pull force. B shows strength vs. depth in the lithosphere for each flow law implemented in this study, assuming a strain rate of $10^{-15}\ s^{-1}$. Brittle failure envelopes for $\phi= 20°$ and $\phi= 10°$ are plotted in red, and diffusion and dislocation strength are in green and blue.

The initial temperature field in the slab tip follows the analytical model from Davies (1999) for a subduction velocity of 12 mm/yr, but the temperature field quickly adjusts as the experiment progresses.

 The rheological properties of the slab are varied to reproduce behavior that is: A–linear viscoplastic, B-viscoplastic following
130 a wet olivine diffusion creep law from Hirth and Kohlstedt (2003), C–viscoplastic following a wet olivine dislocation flow law (Hirth and Kohlstedt, 2003), and D–viscoplastic with diffusion and dislocation creep implemented in parallel (Table 1). We also modified model D in several ways to explore the effects of reduced slab strength on model behavior. We reduce the viscosity of the slab by: E-lowering grain size from 5 mm to 0.5 mm (moderate grain size reduction), F-lowering grain size to 0.005 mm (extreme grain size reduction), and G-imposing a viscosity cap of $1.3 * 10^{24}$Pas. The viscosity cap model mimics the stiffness
135 of the constant-viscosity slab in model A, but the structure of a creep-governed slab. Strength (half the differential stress) predicted by the diffusion and dislocation flow laws is plotted against depth in Figure 1B, assuming the initial $16.25°$Ckm$^{-1}$ temperature gradient, lithostatic pressure, and a strain rate of $10^{-15}$ s$^{-1}$. To implement two laws in parallel, the strain rates predicted by each law are added. The strengths of the laws can vary by several orders of magnitude, so one law typically dominates deformation at a time. The angle of internal friction is 20° in undeformed mantle lithosphere, and weakens linearly
140 to 10° between strains of 0.5 and 1.5. The cohesion is 20 MPa. The plastic and viscous laws are active one at a time, such that only the weaker law controls the effective viscosity of the slab.

 We do not model elastic deformation. Elastic behavior is important when the Deborah number–the ratio of relaxation time to observation time–is high (Reiner, 1964). For representative lithospheric values of viscosity ($\nu = 10^{23}$ Pas) and Young's Modulus ($E = 40$ GPa), the Maxwell relaxation time ($2\nu/E$) is about 160,000 yrs (Turcotte and Schubert, 2014). Our experiments
145 span tens of millions of years, so a viscoplastic approximation is sufficient to capture behavior of interest.

**Table 1.** Parameters for diffusion and dislocation flow laws from experiments on wet olivine aggregates (Hirth and Kohlstedt, 2003). The value of $A$ reported here has been scaled (Ranalli, 1995) to relate the second invariants of stress and strain rate tensors, as used in SULEC, rather than relating uniaxial stress to strain rate, as recorded in the original experiments.

| Parameter | Symbol | Diffusion creep | Dislocation creep |
|---|---|---|---|
| Pre-exponential factor | $A$ | $1.5 * 10^{-18}$ | $5.33 * 10^{-19}$ |
| Power law exponent | $n$ | 1 | 3.5 |
| Grain size | $d$ (m) | 0.005 | – |
| Grain size exponent | $p$ | 3 | 0 |
| Activation Energy | $E^*$ (kJ/mol) | 335 | 480 |
| Activation volume | $V^*$ (m$^3$/mol) | $4 * 10^{-6}$ | $11 * 10^{-6}$ |

**Table 2.** Parameters used for each material in all experiments.

| Parameter | Crust | Lithospheric mantle | Asthenosphere |
|---|---|---|---|
| Thickness (km) | 8 | 80 | 590 |
| Thermal expansion coefficient (K$^{-1}$) | $2.4 * 10^{-5}$ | $2.4 * 10^{-5}$ | 0 |
| Density at $T_0$ (kgm$^{-3}$) | 3200 | 3200 | 3200 |
| $T_0$ ($C$) | 1474 | 1474 | 1474 |
| Thermal conductivity (Wm$^{-1}$K$^{-1}$) | 2.5 | 2.5 | 135.42 |
| Heat capacity (JK$^{-1}$) | 750 | 750 | 750 |
| Viscosity (Pas) | $10^{20}$ | variable ($10^{20}$-$10^{26}$) | $10^{20}$ |

The diffusion and dislocation flow laws follow the form:

$$\dot{\epsilon} = A\sigma^n d^p e^{\frac{-E^* - PV^*}{RT}}, \tag{1}$$

where $A$ is an empirically determined coefficient, $\sigma$ is the stress, $n$ is the stress exponent, $d$ is the grain size, $p$ is the grain size exponent, $E^*$ is the activation energy, $V^*$ is the activation volume, $P$ is the pressure, $T$ is the temperature, and $R$ is the gas constant. The values of these parameters are shown in Table 1.

The temperatures at the top and bottom boundaries of all models are fixed at 0°C and 1474°C, respectively. This imposes a nearly-constant upwards heat flux of 0.04 Wm$^{-1}$. In the lithosphere, which has a thermal conductivity of 2.5 Wm$^{-1}$K$^{-1}$, a thermal gradient of 16.25° per km is required to maintain this heat flux. The asthenosphere has a thermal gradient of 0.3° per km and an artificially high thermal conductivity of 135.4 Wm$^{-1}$K$^{-1}$, in order to mimic the thermal profile of a vigorously convecting mantle, which is not explicitly simulated. Elevated asthenospheric thermal conductivity is often used in numerical

models of subduction (Pysklywec and Beaumont, 2004; Khabbaz Ghazian and Buiter, 2013; Erdős et al., 2021) to maintain a realistic adiabat and a constant heat flux between the asthenosphere and the overlying lithosphere without requiring time to establish vigorous convection prior to simulating processes of interest. For simplicity, we do not implement shear heating or radioactive heat production.

All models simulate a linear viscous asthenosphere, with a viscosity of $10^{20}$ Pas and a constant density of 3200 kgm$^{-3}$ (Fig. 1). The crust on the subducting slab has a constant viscosity of $10^{20}$ Pas. The overriding plate has no crust. The properties of the asthenosphere and thermal properties of the lithosphere remain constant throughout all models (Table 2). The lithospheric density is equal to that of the asthenosphere at a temperature of 1474°C ($\rho_0 = 3200$ kgm$^{-3}$), and increases as temperature decreases according to:

$$\rho(T) = \rho_0 + \rho_0 * \alpha * (T_0 - T),$$    (2)

where $\alpha$ is the coefficient of thermal expansion, $2.4 * 10^{-5}$ K$^{-1}$.

This produces an average density contrast of 63.3 kgm$^{-3}$ between the asthenosphere and the un-subducted portions of the slab. We do not implement any mineral phase changes.

We simulate only the upper mantle, down to 660 km depth. The bottom of the model is a proxy for the mantle transition

zone, where a viscosity increase of one or two orders of magnitude often hinders the sinking of subducting slabs (Hager, 1984). The true mantle transition zone is not impenetrable, making the free-slip model bottom an imperfect approximation (Billen, 2008). The benefit to limiting the model domain rather than simulating a viscosity increase and a lower mantle is that these models are relatively computationally cheap.

## 3    Results

### 3.1    Subduction dynamics

The subduction of the constant $10^{23}$ Pas viscosity slab is illustrated in Fig. 2 A and in the top row of Figure 3. The slab sinks with an increasing velocity until it makes contact with the bottom of the model —a proxy for the viscosity increase at the mantle transition zone. The slab then unbends and flattens out along the bottom of the model, reaching an approximately constant subduction velocity around 11 cm/yr (Fig. 4 A). Subduction stops when the trailing end of the subducting lithosphere,

which has no crust, reaches the trench and stalls subduction by removing weak material from the interface.

The behavior of the three primary creep-governed models contrasts considerably with the behavior of the constant-viscosity model. All three models undergo a similar evolution, illustrated in Figure 2, in which the slab sinks, collides nearly orthogonally with the bottom of the model, and gradually curls under itself. Subduction velocities in all three creep-governed models surpass 20 cm/yr and show no indication of stabilization before the crustless trailing end of the slab jams the trench. Convection at

these later stages is concentrated below the slab, in contrast to the evenly-distributed convection in the reference model (Fig. 3).

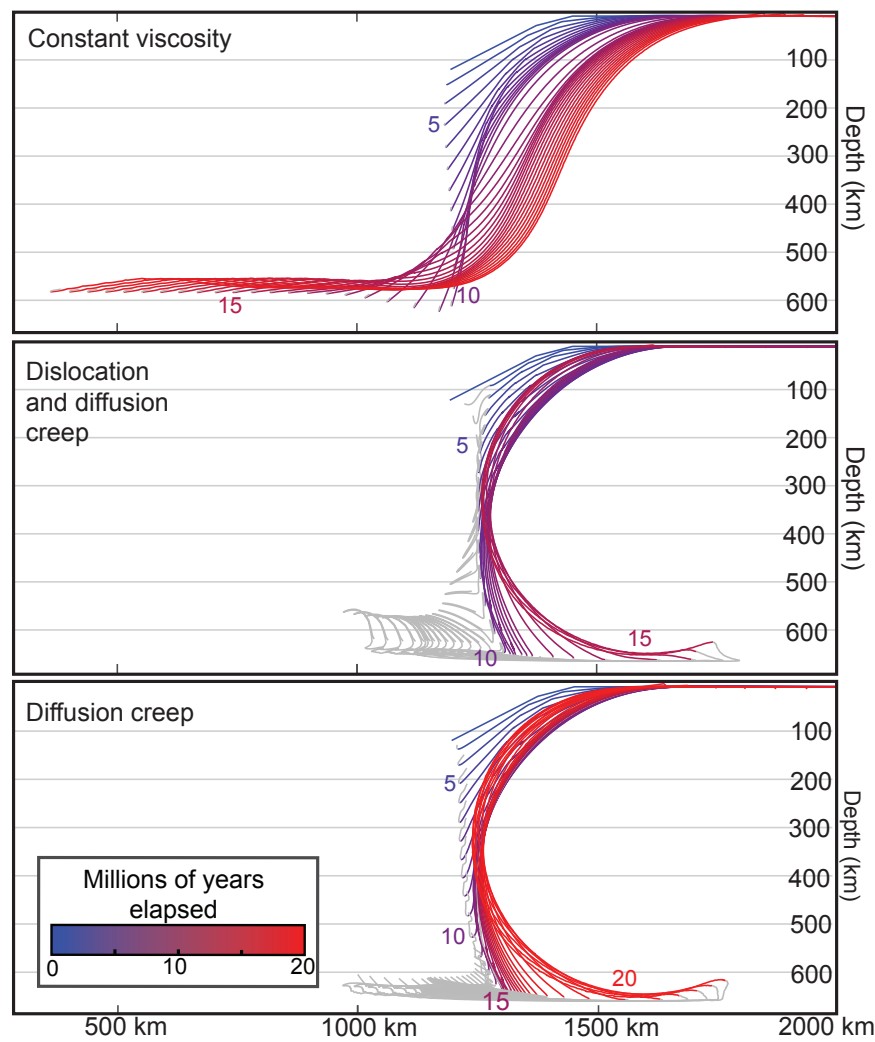

**Figure 2.** Lines show the interface between the crust and the mantle lithosphere in the subducting slab, colored by time from blue to red. Lines are plotted every 200 time steps. Numbers indicating elapsed time in millions of years are located next to the tips of lines at 5 million year intervals. The top plot shows results from the reference model with a constant-viscosity slab. The middle plot shows the model with dislocation and diffusion creep implemented in parallel, which is almost indistinguishable from the model with only dislocation creep (not shown). The lowest plot shows the model following only a diffusion creep law. Gray lines trace portions of the slab that have reached asthenospheric viscosity ($10^{20}$ Pas).

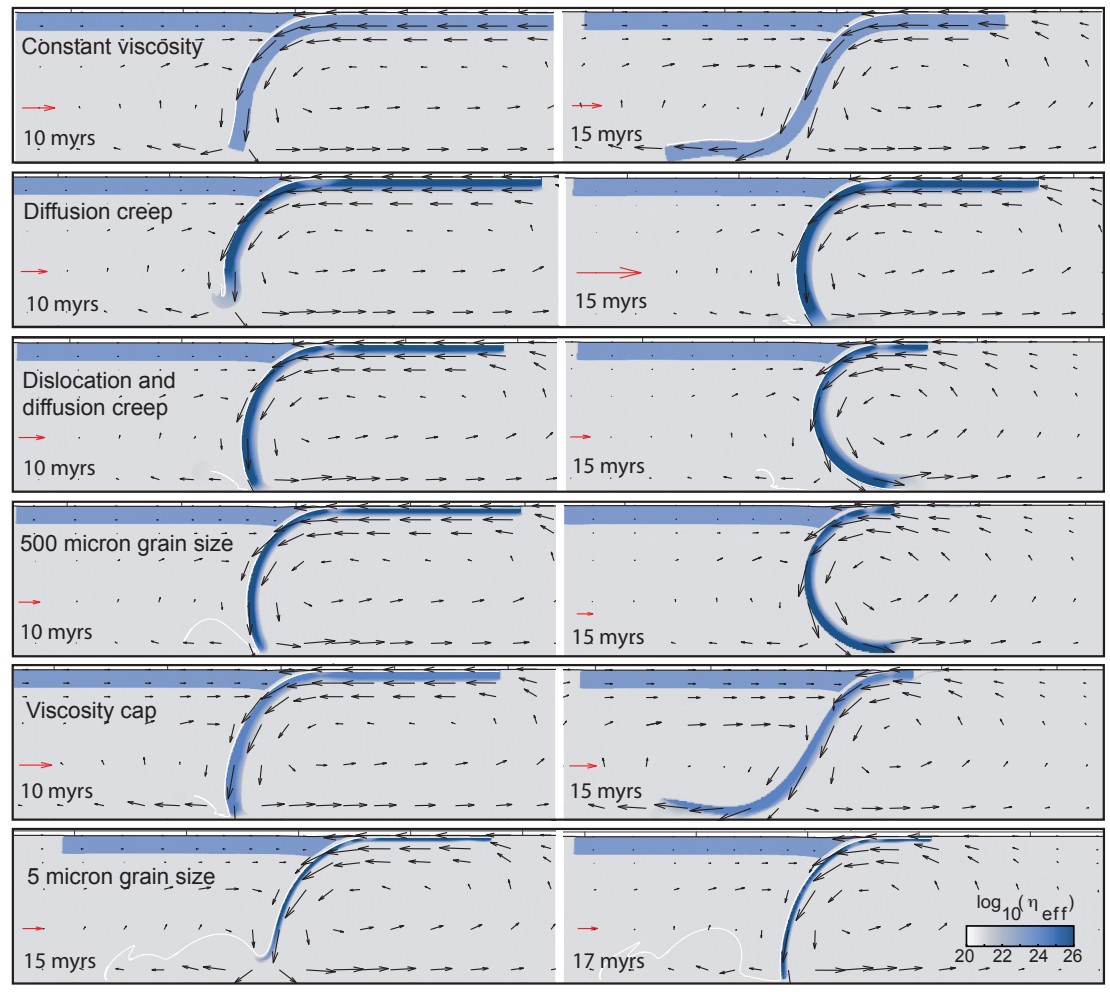

**Figure 3.** Slabs at 10 and 15 million years, colored by viscosity, with vectors showing the velocity field. White lines indicate material at the crust-mantle interface in the subducting plate. The plots for dislocation-only and dislocation-diffusion models are indistinguishable, so we only show results from the dislocation-diffusion model. The model with a 5 micron grain size is shown at 15 myrs and 17 myrs to show the late-stage morphology. The velocity arrows have different scales in each panel because the maximum velocity varies between the snapshots presented. Each red arrow is scaled to 10 cm/yr for the given panel. Plots use the oslo colormap from Crameri (2018).

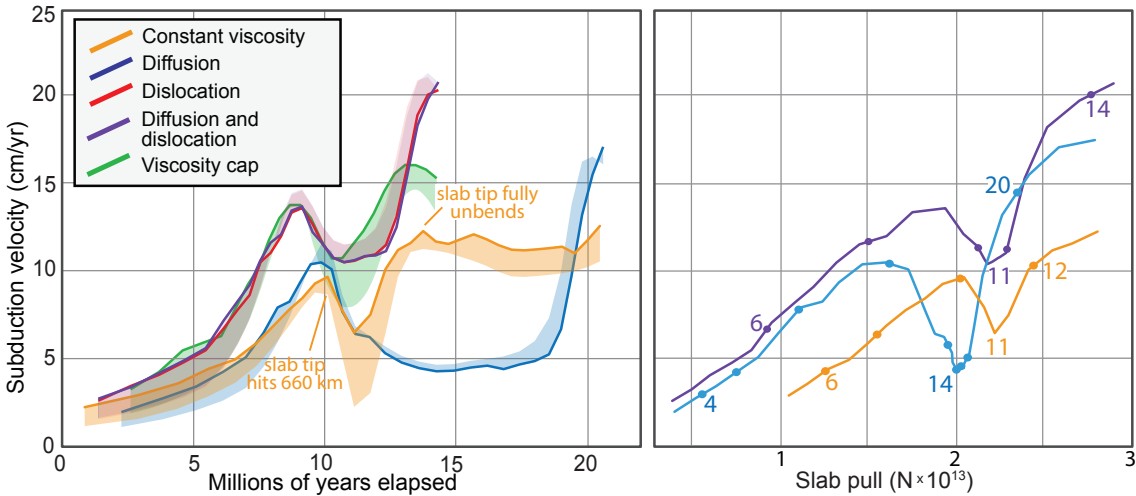

**Figure 4.** The plot on the left shows subduction velocity over time for the constant-viscosity reference slab and four creep-governed slabs. The bold curves show the rate at which material is consumed at the trench and the opposite edge of the shaded region around each curve shows the rate of motion of the slab with respect to the model boundaries. The thickness of the shaded region therefore represents the velocity of the trench. During periods where the shaded region lies below the bold line, the model undergoes slab rollback. Where the shaded region lies above the line, there is trench advance. The plot on the right shows how subduction velocity varies with slab pull in the constant-viscosity, diffusion, and diffusion-dislocation models. The lines follow the progression of each model through time with points at 2 million year intervals. Data are only plotted until the slab begins to lie flat on the bottom of the model. Beyond this point, slab pull is no longer proportional to the density anomaly of the subducted slab, and the calculation of slab pull is less straightforward.

The shaded regions in Figure 4 represent the rate of trench rollback/advance over time. In all models, subduction takes place via trench rollback and advance of the un-subducted plate as the tips of the slabs fall freely through the asthenosphere. Trench rollback plays a proportionally smaller roll as subduction velocity increases. The constant-viscosity slab remains in trench retreat throughout the experiment, but the creep-governed slabs switch to trench advance as they approach the mantle transition zone. This difference is a consequence of the closed "fish tank" form of our models. After the slabs make contact with the bottom of the model, the lithosphere prevents asthenosphere from flowing between the left and right sides of the model. This causes trench rollback velocity to be linked to subduction velocity, as pointed out by Billen (2008). In the constant-viscosity model, the trench moves rightward to compensate for the leftward motion of the lithosphere. In each creep-governed model, as the slab curls under itself, the rightward motion of the slab tip is balanced by trench advance to avoid compression of the material on the right side of the model. The thickness of the lithosphere is approximately one eighth of the model thickness, so the observed trench rollback/advance speeds are a comparable proportion of the subduction velocity.

## 3.2 Slab viscosity structure

The cores of the creep-governed slabs exceed the viscosity of the constant $10^{23}$ Pas reference model by several orders of magnitude. The diffusion-only slab has the highest viscosity overall. At the surface of the model, in areas with no active plastic deformation, the slab reaches the viscosity cap of $10^{26}$ Pas in the top 40 km (Fig. 5 Profile C). The viscosity decreases between 40 and 75 km depth according to the diffusion creep law, and hits the viscosity minimum of $10^{20}$ Pas below 75 km. The viscosity structures of the dislocation-only and dislocation-diffusion slabs are extremely similar, suggesting that dislocation creep dominates in the upper mantle when diffusion and dislocation creep are implemented simultaneously. The viscosity of the un-subducted portions of slabs with dislocation creep decreases from $10^{26}$ to $10^{20}$ Pas in the 35 to 60 km depth range. Once subducting lithosphere has heated to near-asthenospheric temperatures, its viscosity decreases to the minimum value ($10^{20}$ Pas) and its density approaches asthenospheric density. A growing proportion of the slab tip, represented by the thin gray lines in the lower two plots in Figure 2, therefore assimilates into the mantle as the model progresses.

Figure 5 shows the strength of the dislocation-diffusion slab along two profiles: one in the subducted portion of the slab around 400 km depth and one where the slab bends just before entering the trench. The true strengths—computed as half the modeled differential stress—are compared with the strength envelopes calculated analytically assuming dislocation creep, diffusion creep, and brittle failure under the temperatures, pressures, and strain rates along the profiles. The true strength is controlled by the dislocation creep mechanism in regions of both profiles. In A, the true strength curve deviates from the dislocation creep curve in the top 30 km of the slab where the viscosity reaches the $10^{26}$ Pas stress cap. The slab has a constant viscosity over this interval, but lower strain rates in the core result in lower stresses and a dip in strength centered approximately 20 km into the slab. In profile B, deformation is accommodated by brittle failure in the top 20 km. Diffusion creep plays almost no role in the slab, consistent with the models by Garel et al. (2014), in which diffusion creep is active primarily in the asthenosphere in the upper mantle.

## 4 Force Balance and Energy Dissipation

In this section, we lay out equations describing the resisting and driving forces in our models. We conduct a hybrid force-balance/energy dissipation analysis, using the rate of energy dissipation (which is calculated at each time-step in SULEC) to break down resistance in the crust, lithosphere, and surrounding mantle. We compare the behavior of the creep-governed and reference slabs before and after interaction with the bottom boundaries of the models. Early differences in subduction velocity can be attributed to both lowered bending resistance at the trench of the creep-governed slabs and lower asthenospheric drag around the slab tip. The creep-governed slabs exhibit a more pronounced dip in velocity when they begin to interact with the transition zone, due to their greater bending resistance at depth. Once the creep governed slabs assume a low-resistance overturned geometry, their subduction velocity increases sharply relative to the reference model.

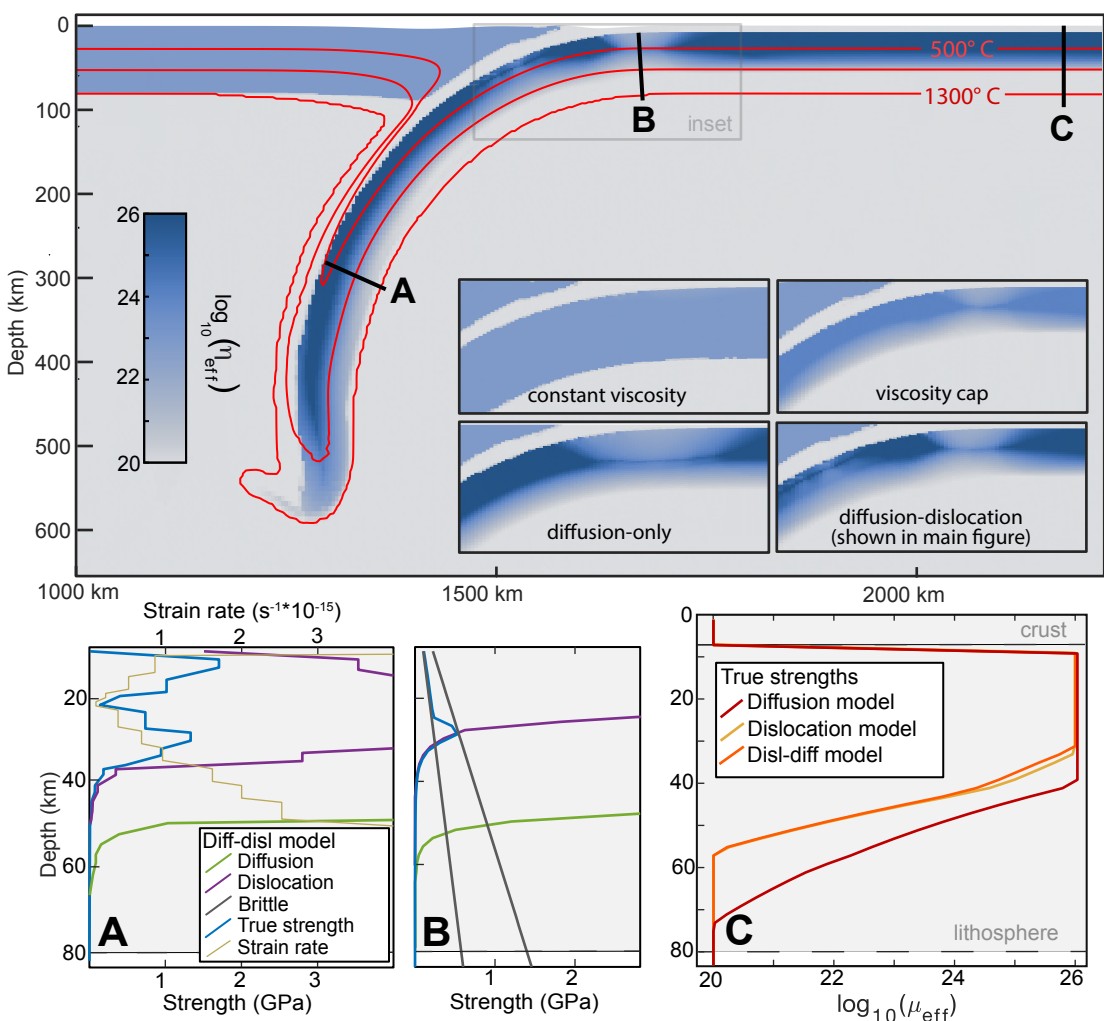

**Figure 5.** An image of the viscosity structure of a slab following dislocation and diffusion creep laws after 10 million years. Plots along profiles A and B show the strength of the slab (blue), compared to the strengths predicted by dislocation creep (purple), diffusion creep (green), and brittle failure (gray) with $\phi = 20°$ and $\phi = 10°$ under local strain rates and pressures. Profile A shows strength at middle depths of the upper mantle, where the upper half of the slab hits the viscosity cap and no brittle deformation is active. The thin yellow line shows strain rate (top axis), which reaches a minimum approximately 20 km into the slab. Plot B shows strength at the surface near the trench where the slab bends and undergoes brittle deformation. Plots in C show viscosity vs. depth in the diffusion, dislocation, and diffusion-dislocation slabs. These plots represent un-subducted lithosphere with strain rates too low for plastic deformation. The location of these profiles is indicated on the model image. The insets show the viscosity structures of slabs where they bend at the trench. Each snapshot is taken from a similar point in the subduction process, with the slab tip approximately 100 km from the bottom boundary. All models have some plastic failure in this region, but the degree to which plastic failure lowers bending resistance is much higher in the creep-governed models than in the constant-viscosity model.

## 4.1 Driving and Resisting Forces

Subduction is driven by gravitational force on the dense lithospheric material at the surface of the model. This "slab pull" force is approximated as:

$$F_{sp} = g \int_{V \, slab} (\rho_{lith} - \rho_{asth}) dV,\tag{3}$$

where $\rho_{lith}$ is the temperature dependent density of the lithosphere, $g$ is the gravitational acceleration, and $\rho_{asth}$ is a constant: $3200 \, kg/m^3$. The density contrast is integrated over all lithospheric material that has passed through the trench into the subduction zone. Slab pull evolves as subduction progresses according to the amount and temperature of the subducted lithosphere, which are functions of subduction velocity. Subduction velocity is in turn determined by the balance between slab pull and resisting forces in the lithosphere and asthenosphere.

We use the rate of energy dissipation in our models–which is automatically output by SULEC– to understand the resisting forces at play. The rate of internal dissipation of energy is calculated as:

$$\dot{W} = \int_{V} \sigma_{\mathrm{II}} \dot{\epsilon}_{\mathrm{II}} dV,\tag{4}$$

where $\sigma_{\mathrm{II}}$ and $\dot{\epsilon}_{\mathrm{II}}$ are the second invariants of the stress and strain tensors, respectively, at a particular point in the model. Their product is integrated over the volume of interest (e.g. the crust). This quantity is a rate of work, proportional to the forces and the velocities throughout the volume.

Resisting forces in our models can be categorized into 1) forces acting along the interface of the subducting and overriding plates, 2) resistance to slab bending, 3) resistance to slab stretching, and 4) viscous drag in the asthenosphere. Resistance at the plate interface is primarily reflected by energy dissipated in the crust of the subducting slab (Fig. 7A). Energy dissipated in the surrounding mantle reflects viscous drag on the plate (Fig. 7B), and energy dissipated in the plate itself reflects deformation of the slab (Fig. 7C).

The rate at which potential energy is dissipated depends on the rate at which the subducting slab moves dense material down through the model. Therefore, subduction velocity has a strong correlation with overall energy dissipation rate (Fig. 6). Comparison of energy dissipation rate in each material at a given subduction velocity allows for comparison of resisting force in the crust vs. asthenosphere vs. lithosphere between models, controlling mostly for overall energy dissipation rate (Fig. 7). For instance, the lithosphere provides less resistance, and the crust and asthenosphere more resistance at high subduction velocities in the diffusion-dislocation model relative to the diffusion-only model.

Energy dissipation rate in the lithosphere does not have a straightforward relationship with subduction velocity (Fig. 7 C). Asthenospheric energy dissipation rate, on the other hand, has a strong dependence on subduction velocity (Fig. 7B). Therefore, lower resistance to deformation in the slab allows more energy to be dissipated in the asthenosphere and results in a higher subduction velocity.

Lithospheric energy dissipation can be broken down into bending and stretching components. Ribe (2001) outlines the relationship between $M$, the bending moment on a thin viscous sheet, $N$, the stretching moment (the integral of slab-parallel

**Table 3.** The approximate resistance to bending at the trench and at depth in five of the models presented. Measurements are made once the final geometry has been established. This occurs after different amounts of elapsed time in each model, so time of measurement is listed in column 2. Note the correlation between stiffness at the trench and initial subduction velocity, and the correlation between stiffness at depth and the final geometry of the slab (Fig. 3). *Values marked with asterisks are less precise due to low bending rates (Appendix A)).

| Model | Time elapsed (myrs) | Stiffness at trench ($Pasm^3$) | Stiffness at depth ($Pasm^3$) | Geometry |
|---|---|---|---|---|
| Reference ($10^{23}Pas$) | 15 | $10^{36} - 10^{37}$ | $10^{37}$ | Feeds forward |
| Diffusion-only | 20 | $10^{36} - 10^{37}$ | $10^{37} - 10^{38}*$ | Curls under |
| Dislocation-diffusion | 14 | $10^{35}$ | $10^{38}*$ | Curls under |
| Reduced grain size (500 micron) | 14 | $10^{35}$ | $10^{37} - 10^{38}*$ | Curls under |
| Viscosity cap | 15 | $10^{35}$ | $10^{37}$ | Feeds forward |

stresses), and the rates at which the sheet (slab) bends and stretches:

$$\begin{bmatrix} N \\ M \end{bmatrix} = \begin{bmatrix} 4\mu H & 5\mu H^3 K/6 \\ \mu H^3 K/3 & \mu H^3/3 \end{bmatrix} \begin{bmatrix} \Delta \\ \Omega \end{bmatrix}. \tag{5}$$

Here, $\Delta$ is the rate of stretching, $\Omega$ is the rate of bending, $H$ is the thickness of the sheet, $K$ is the curvature, and $\mu$ is the viscosity of the sheet (assumed constant). The quantity $\mu H^3/3$ is the resistance to bending, which we will refer to as $D$.

In models presented by Capitanio et al. (2007), bending plays a larger role in the dissipation of energy than stretching in stiff

slabs, accounting for >80% of the total dissipation. All of the slabs in our models can be considered stiff by these standards, and stretch minimally. Therefore, we discuss only bending resistance. The lower row in Equation 5 can be rearranged as:

$$D = M/(K\Delta + \Omega). \tag{6}$$

In models where viscosity changes as a function of depth within a slab, $\mu H^3/3$ cannot be used to calculate a slab's resistance to bending. However, we can still think about bending resistance as the relationship between bending rate and moment expressed

in Equation 6. Therefore, we calculate bending moment, bending rate, stretching rate, and curvature in our slabs and use Equation 6 to estimate slab stiffness (Appendix A). It is worth noting that this calculation relies on smoothing of rough, discrete measurements of curvature and velocity along the length of the slab and is therefore only approximate. Still, the calculation provides insight into how resistance to bending changes with various implementations of slab rheology.

We measure bending resistance along two profiles in each slab: near the trench, where plastic failure reduces the effective

slab thickness, and at the bend near the mantle transition zone, where effective viscosity is determined entirely by ductile creep and the viscosity cap (Table 3). We measure at points along the slab where the bending rate reaches a local maximum.

### 4.2 Energy Dissipation Over Time

Figure 4B shows the relationship between slab pull and subduction velocity over time in each model. The dislocation-diffusion and dislocation-only slabs follow very similar paths, so the data for the dislocation-only slab are not shown. At a given slab

pull, the dislocation-governed slabs have a higher subduction velocity than the diffusion-governed slab, which has a higher subduction velocity than the constant-viscosity slab (Fig. 4B). This implies that the constant-viscosity model experiences the highest cumulative resisting force at a given subduction velocity.

There is an approximately quadratic relationship between velocity and dissipation rate in the linear viscous crust and asthenosphere (Fig. 7). The rate of energy dissipation in an area is proportional to local strain rate and stress. Stress in the linear
viscous materials is again proportional to strain rate. Therefore, strain rate (and by extension subduction velocity) enters into the equation for dissipation rate twice: once directly and once in the stress term, resulting in a quadratic. This relationship is complicated by the geometry of the model. For instance, the crust deforms not only along the interface between the subducting and overriding plates, but also along a constantly increasing length of subducted lithosphere.

In the first ~10 million years of our experiments, the geometry of subduction is similar in all models, and subduction velocity
varies subtly due to differences in the bending resistance at the trench and drag in the asthenosphere. Once the slabs contact the bottom boundaries, the model geometries diverge and subduction rates vary more dramatically.

### 4.2.1 Energy Dissipation Before Contact with the Transition Zone

From the start of each experiment until the slabs reach 660 km depth, the rates of energy dissipation in the asthenosphere and lithosphere are highest in the reference model and lowest in the models implementing dislocation creep, resulting in a lower
initial subduction velocity in the reference model. The rate of energy dissipation in the crust is very similar between models at a given subduction velocity (Fig. 7).

The constant-viscosity slab subducts more slowly than the creep-governed slabs despite much higher viscosity throughout most of the creep-governed slabs. Two factors likely explain this pattern: (1) the bending resistance of the creep-governed slabs is more dramatically reduced by plastic failure at the trench because the effective thickness of the slabs is lower, and (2) drag
around the tip of the creep-governed slabs is lower, although the first factor likely plays a much larger role. The insets in Figure 5 illustrate how much more dramatically plastic failure reduces viscosity at the trench of the creep-governed models.

We calculate resistance to bending at the trench to estimate the extent of this plastic weakening across our models. For the details of this calculation, see Appendix A. We find that the reference slab has a bending resistance of $10^{36}$-$10^{37}$ $\mathrm{Pasm}^3$, only slightly below that expected based on the thickness and viscosity of the slab ($1.2 * 10^{37}\mathrm{Pasm}^3$). In contrast, the slabs
implementing dislocation creep (including the slab with a viscosity cap) appear to have a bending resistance on the order of $10^{35}\mathrm{Pasm}^3$. The model with grain size reduced by an order of magnitude (to 0.5 mm) has a very similar structure and bending resistance to the model with 5 mm grain size (Fig. 3). We calculate the bending resistance of the diffusion-only slab as $10^{36}$-$10^{37}\mathrm{Pasm}^3$ near the trench; substantially higher than that of the dislocation-governed slabs due to the greater effective thickness of the diffusion-only slab. This contrast in bending resistance at the trench contributes to the lower subduction velocities of the
diffusion-only and reference slabs early in the experiment.

The rate of energy dissipation in the asthenosphere is higher at a given subduction velocity in the reference model than in the creep-governed models (Fig. 7). This indicates that the contrast in bending resistance at the trench may not be solely responsible for the difference in slab velocity. Subduction in the reference model may be further slowed by the slab tip, which

remains at $10^{23}$ Pas in the reference model and assimilates into the $10^{20}$ Pas mantle in the other models. Panels D and G of Figure 7 illustrate this contrast: at the time steps shown, the slab pull force is about $2*10^{13}N$ in both models, but the viscous portion of the constant-viscosity slab is longer than that of the diffusion-dislocation slab. The warm parts of the slab do not contribute to slab pull in any of the models as their density is equal to the surrounding mantle, but the strong slab tip in the reference model may increase drag on the reference slab, slowing subduction.

It is worth noting that the warm areas of the lithosphere behave as asthenosphere in the creep-governed models but are still counted towards the total lithospheric energy dissipation rate in Figure 7, which could account for some of the discrepancy between the asthenospheric energy dissipation rates in the reference and creep governed models. Indeed, according to the scaling analysis by Ribe (2010), we would expect the subduction velocity of our slabs to be primarily governed by the viscosity structure of the slab, rather than by drag in the asthenosphere. Slabs with a dimensionless stiffness greater than 1 should sink with a velocity controlled primarily by the slab bending resistance. Ribe defines stiffness as: $S = \gamma(H/l_b)^3$, where $\gamma$ is the slab-mantle viscosity contrast and $l_b$ is the length over which the slab bends. Our slabs bend over a section on the order of 100 km long and have a viscosity contrast of $10^6$, implying that they have a stiffness well above 1, and asthenospheric drag should be a secondary control on subduction velocity.

The model with a reduced viscosity cap ($1.3*10^{24}$Pas) behaves in a very similar manner to the diffusion-dislocation slab until the slab hits the bottom of the model (Fig. 4). The stiffness at the trench is similar in both models, despite the difference in maximum viscosity, because the effective thickness of the plates is determined by the same creep laws, and plastic failure is active throughout the stiff portion of both plates, reducing viscosity far below the cap. In contrast, the effective thickness of the diffusion-only slab is greater, increasing the bending resistance at the trench and slowing subduction at the earlier stages (Fig. 4). Similar bending resistance at the trench and similar drag on the slab tip between the viscosity cap and diffusion-dislocation models cause the reduced viscosity cap to have almost no effect on subduction velocity as the slab falls freely through the mantle.

### 4.2.2   Energy Dissipation After Contact with the Transition Zone

Once plates have made contact with the bottoms of their respective model domains, subduction velocity varies dramatically between models. Most of this difference can be attributed to the geometries adopted by the plates (Fig. 6).

All slabs slow as they come into contact with the bottom of the model. This reduction in velocity is most pronounced in the creep governed slabs, which have a higher bending resistance in the lower mantle than the reference slab (Table 3), and deform more slowly at the boundary. However, once the creep-governed slabs become overturned, their subduction velocity increases dramatically since they are able to maintain an approximately constant curvature after bending at the trench (Fig. 7I), whereas the constant-viscosity slab must bend again to feed forward at the mantle transition zone. The free-slip boundary at the bottom of the models likely exacerbates this acceleration, as other implementations of the transition zone might provide more resistance to lateral sliding of the slab at 660 km.

After the constant-viscosity slab develops a second bend at the bottom of the model (Fig. 7F), the overall resisting forces in the model reach an equilibrium with slab pull, resulting in a constant subduction velocity and a constant rate of energy

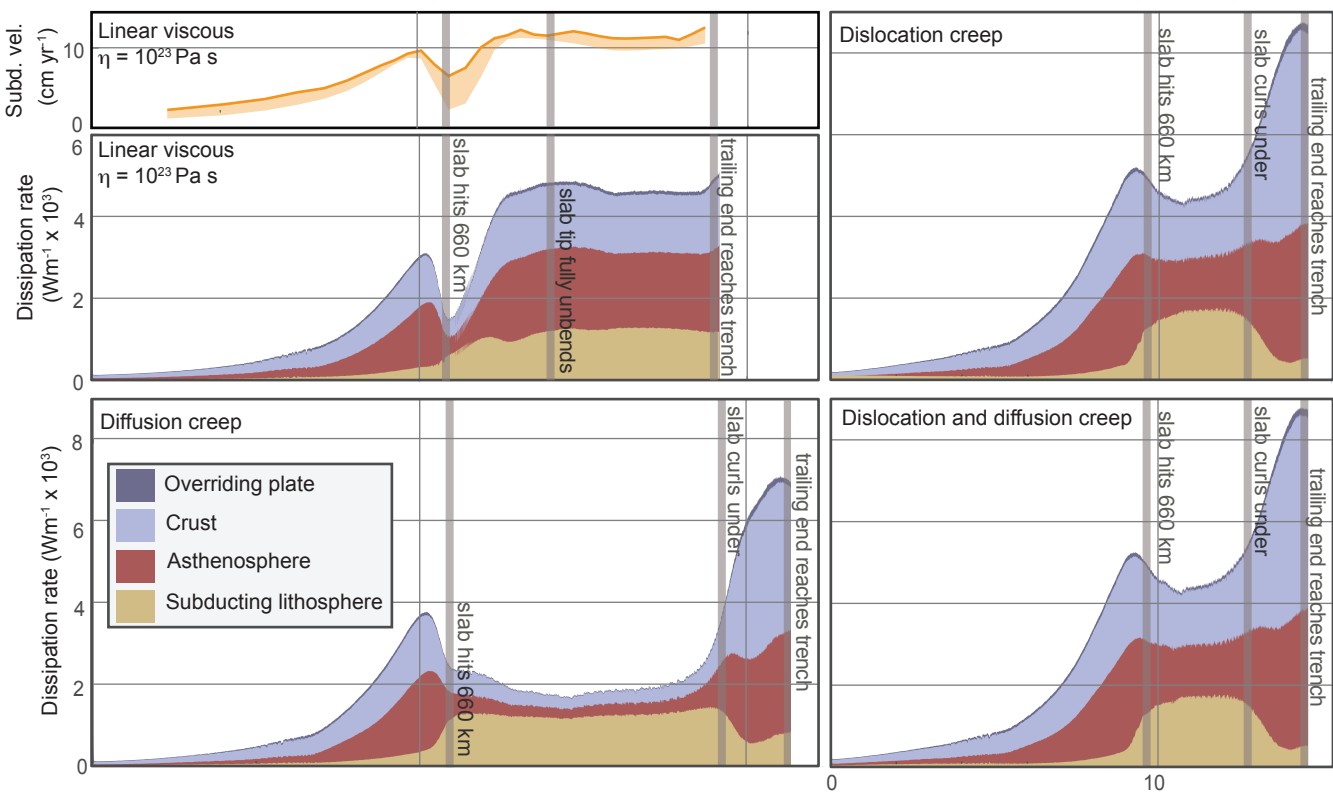

**Figure 6.** Dissipation of energy over time in each model. Each curve is broken down into dissipation in the crust, mantle lithosphere, asthenosphere, and the overriding plate. The plot on the upper left shows subduction velocity for the reference model to illustrate the correlation with the total rate of energy dissipation shown in the plot below. Vertical grey bars mark notable points in the evolution of each experiment.

dissipation (Fig. 4A). This stage is also observed in the analogue models by Funiciello et al. (2008). In contrast, lithospheric dissipation rate in the creep-governed slabs decreases as the plates overturn, reflecting the decrease in bending resistance once the slabs curl under. This decrease in lithospheric dissipation rate is marked with vertical gray lines in Figure 6, and coincides with a rapid increase in subduction velocity. Given an infinitely long slab, subduction velocity would likely eventually level off at some higher rate once the resistance in the mantle and crust became high enough to balance slab pull. However, the creep-governed models are not able to reach equilibrium between driving and resisting forces before the slab is consumed, so subduction velocity continues to increase until the end of the experiments.

Capitanio et al. (2009) find that the partitioning of energy dissipation into the lithosphere is lower in layered slabs than in constant-viscosity slabs (~20% vs. ~40%), even when all slabs have the same deflected geometry. This effect could play a role in the low lithospheric energy dissipation rate in creep-governed slabs late in our experiments. However, the difference

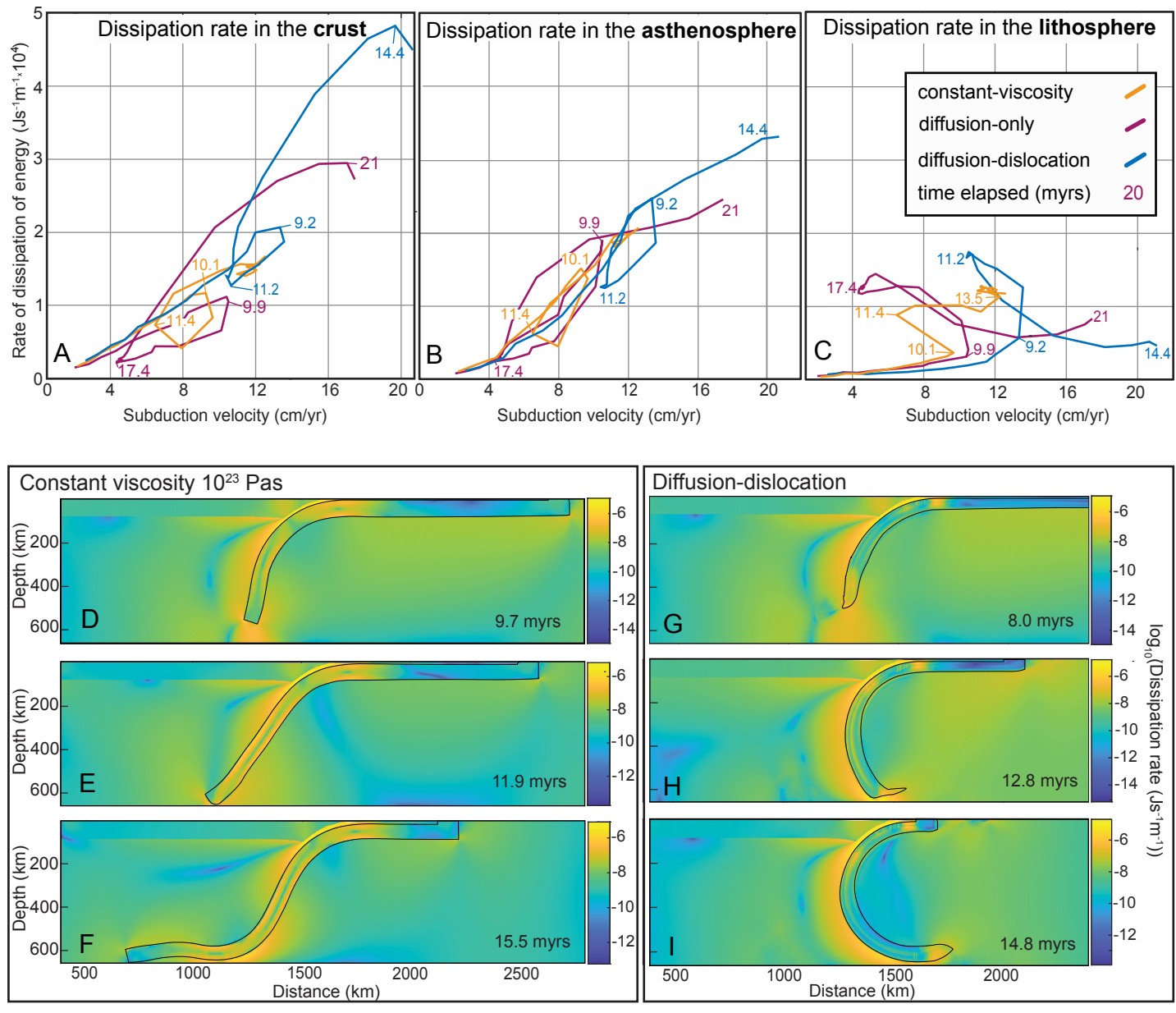

**Figure 7.** Dissipation rate in the crust (A), sub-lithospheric mantle (B), and slab mantle lithosphere (C) plotted against subduction velocity. The lines trace out the progression of each model over time, with selected points in time indicated in millions of years. Panels D-F show the distribution of energy dissipation in the reference model at 3 key snapshots. Panels G-I show distribution of energy dissipation for similar stages in the parallel diffusion-dislocation model. Plots are colored according to $log_{10}$ of dissipation rate in J s$^{-1}$ m$^{-1}$. In each panel, the portion of the slab that exceeds the viscosity of the asthenosphere is outlined in black.

in lithospheric energy dissipation rate between our creep-governed and constant-viscosity models likely primarily reflects the contrast in slab morphology rather than the viscosity structure of the slabs.

Though the slab with a lower viscosity cap behaves almost identically to the diffusion-dislocation slab at early stages, after reaching the bottom of the model, the reduced stiffness at depth relative to the diffusion-dislocation slab allows the viscosity-cap slab to feed forward like the constant-viscosity slab. The subduction velocity of the viscosity-cap slab therefore does not increase dramatically in the later stages of subduction like that of the diffusion-dislocation slab (Fig. 4), since it is not able to maintain constant curvature. The viscosity-cap slab does, however, subduct more quickly than the reference slab after contact with the transition zone, since the bending resistance at the trench is lower than that of the thicker reference slab.

Differences in subduction velocity and model evolution between models primarily reflect differences in resistance within slabs, as discussed above. This is unsurprising because slab rheology is the only parameter that varies between models. However, as the model geometries diverge, subtle differences emerge in the energy dissipation rates in the crust and sub-lithospheric mantle as well.

Dissipation rate in the crust has a similar relationship to subduction velocity in all models until the tip of the slab begins to interact with the bottom of the model domain around 9-10 myrs (Fig. 7). When the slab curls under after 17 myrs in the dislocation-governed models and 11 myrs in the diffusion-governed model, dissipation rate in the crust increases dramatically compared to the earlier stages of subduction, primarily due to high dissipation in the crust dragged between the plate and the bottom of the model (Fig. 7H). This effect is not present in the constant-viscosity model where the slab does not curl under and the crust never contacts model boundary.

Dissipation in the asthenosphere shows a similar dependence on subduction velocity across all models (Fig. 7), even though convection is concentrated underneath the subducting plate in the creep-governed models and evenly distributed in the constant-viscosity slab model (Fig. 3). It is important to note that the rheology of the asthenosphere is highly simplified, and the relationship between subduction velocity and asthenospheric dissipation rate may not remain similar between models if a more realistic rheology were implemented. Asthenospheric energy dissipation rate is slightly lower at high subduction velocities in the creep-governed models than the extrapolated pattern from the reference model would suggest (Fig. 7A), likely because the high-dissipation region between the slab and the bottom of the model is crust in the creep-governed models and asthenosphere in the reference model.

## 5 Discussion

### 5.1 Comparison to realistic slab rheology

Empirical evidence for the viscosity of subducting lithosphere is limited, but generally suggests weaker behavior than exhibited by the creep-governed models presented here. Several studies have determined realistic slab-to-mantle viscosity ratios by comparing the results of numerical or analogue models to observed slab geometries or the geoid. Three dimensional whole-Earth numerical models with a viscosity contrast between $10^2$ and $10^4$ produce the most realistic plate behaviour (Zhong et al., 2008). Similar models run by Mao and Zhong (2021) best fit the geoid with contrasts between $10^1$ and $10^2$. Ribe (2010)

matches observed slab curvatures with slab-to-mantle viscosity ratios of 140-510, and Loiselet et al. (2009) obtain realistic slab curvature with a ratio of 45. The analogue models of Schellart (2008) fit true slab geometries best with a viscosity ratio of 100-700, and Conrad and Hager (1999) propose a ratio of 50-200. These studies suggest that, for a sub-lithospheric mantle viscosity of $10^{19} - 10^{20}$ Pas, overall slab viscosity should not exceed $10^{23}$ Pas. Seismic strain rates from the Pacific plate at the Tonga-Kermadec trench (Holt, 1995) also support absolute viscosity around $10^{23}$ Pas within the slab (Billen et al., 2003). This is three orders of magnitude lower than the highest viscosity in our models, which is already restricted to $10^{26}$ Pas by the maximum viscosity cap imposed by computational constraints. It is worth noting that all of the studies cited here model slabs with a constant viscosity, and therefore do not preclude higher viscosity in small portions of a heterogeneous subducting plate.

The slab morphologies in our models provide further evidence that plates controlled exclusively by diffusion and dislocation creep have unrealistically high bending resistance. Most modern subducting slabs flatten out forward between 500 and 1200 km depth, or sink at a steep angle through the mantle transition zone (Goes et al., 2017), whereas the creep-governed slabs in our models overturn at the transition zone. The only slab observed to curl under itself, as our three creep-governed slabs do, is the Indian plate beneath the Himalayas (Goes et al., 2017). This plate also had a very high subduction velocity, like our creep-governed slabs, though the apparent similarities to our models are likely a coincidence since some authors (Qayyum et al., 2022) argue that the Indian slab overturned recently due to a period of trench advance, rather than maintaining an overturned geometry for an extended period.

It should be noted here that our models approximate the base of the upper mantle as a hard boundary, which undoubtedly has an impact on slab morphology and subduction dynamics once the slab tip reaches the bottom of the model. For this reason, divergence from realistic behavior at the later stages of our experiments cannot be entirely attributed to high slab stiffness. If the models presented here had a viscosity contrast at 660 km depth, rather than a hard boundary, the creep-governed slabs may have penetrated the mantle transition zone. Sufficiently stiff slabs in the models of Garel et al. (2014) approach the transition zone bent, like our slabs, but, upon reaching 660 km depth, continue vertically downwards or undergo trench retreat to bend forward. On the other hand, in the 3-dimensional models of Stegman et al. (2010), which also simulate a viscosity contrast at 660 km depth, slabs with sufficiently low Stokes buoyancy, and with comparable stiffness to our creep-governed slabs ($\frac{\eta_{slab}}{\eta_{mantle}}(\frac{H_{slab}}{H_{uppermantle}})^3 = 1 - 100$), overturn, similar to our models. It is difficult to know which morphology our slabs would exhibit if our models had included a lower mantle. Regardless, our overturned, creep-governed slabs appear unusually stiff, despite moderate (<80 km) effective thicknesses.

Although viscosities modeled here appear unrepresentative of most real subduction zones, they do not stand out among numerical models of subduction in the literature. The slabs modeled by Gerya et al. (2021), Khabbaz Ghazian and Buiter (2013), Tetreault and Buiter (2012), Tagawa et al. (2007), Billen and Hirth (2007), and Erdős et al. (2021), among others, reach $10^{25}$ or $10^{26}$ Pas in regions tens of kilometers thick. In shallow models, the lithosphere tends to curl fully underneath itself (Khabbaz Ghazian and Buiter, 2013), as we observe in our creep-governed models. And in deeper models (Tagawa et al., 2007; Billen and Hirth, 2007), slabs often retain curvature through the mantle transition zone (Billen, 2008).

The high bending resistance of our creep-governed slabs supports previous findings (Kameyama et al., 1999; Čížková et al., 2002; Garel et al., 2014), that weakening mechanisms that we have not implemented play an important role in subduction zone

deformation. Karato et al. (2001) and Kameyama et al. (1999) discuss the importance of Peierls creep to deformation in the cold interiors of subducting plates. Kameyama et al. (1999) show that, for grain sizes of approximately 1 mm, Peierls creep is active above stresses around 1000 MPa. In our models, creep-governed plates reach differential stresses above 1000 MPa in a region several tens of kilometers in thickness along most of their length, implying that Peierls creep should play an important role in their deformation.

Karato et al. (2001) proposed that grain size reduction due to mineral phase changes around 400 km depth could weaken the diffusion creep mechanism, helping to explain the discontinuity between observed slab strength and the predictions of diffusion and dislocation flow laws. Ĉížková et al. (2002) found that simulating a grain size reduction of several orders of magnitude weakens slabs significantly, but plays a smaller role than a stress cap approximating Peierls creep.

Elsewhere, Gerya et al. (2021) argue that brittle failure at the trench may concentrate at periodic intervals along the length of the slab. They show that, near the trench, below regions of extensive brittle deformation, increased stress may cause ductile damage that lowers grain sizes in the center of the slab. As the subducting plate moves deeper into the upper mantle, damaged areas maintain lower viscosities than undeformed areas, leading to sausage-like segmentation of the slab. The segmented slabs in their models move forward at the mantle transition zone despite reaching a viscosity of $10^{25}$ Pas in strong regions.

Our results indicate that, using the wet olivine flow laws from Hirth and Kohlstedt (2003), grain size weakening alone is not sufficient to produce slabs that feed forward at depth. We run three models implementing dislocation and diffusion creep with grain sizes of 5 mm, 0.5 mm, and 0.005 mm throughout the slab. This approach is much less sophisticated than that of Gerya et al. (2021), but illustrates the behavior of a slab with strong grain size weakening but a very high viscosity cap ($10^{26}$ compared to $10^{25}$ in Gerya et al. (2021)). Grain size reduction from 5 mm to 0.5 mm has little effect on the slab's structure or behavior (Fig. 3). The slab with 0.005 mm grain size slab (animations are included in the Supplementary Materials) represents a true end-member case with an extremely fine grain size throughout. The uniformly thinner slab in our models likely behaves differently than a more realistically segmented slab. Still, the fact that the slab does not bend forward at the transition zone (Appendix A) despite its extremely small grain size supports the findings of Ĉížková et al. (2002), who show that, with a high stress limit, slabs are too strong to bend forward at the mantle transition zone, regardless of grain size.

Our analysis shows that bending resistance at the trench influences subduction velocity, while bending resistance at depth influences plate interactions with the transition zone (Table 3). A viscosity or stress-limiting mechanism is required for plates to bend forward at the transition zone. Therefore, after interaction with the transition zone, subduction dynamics become unrealistic in models controlled only by diffusion and dislocation creep, with no stress-limiting mechanism.

However, the maximum viscosity/stress reached in a slab has little impact on the rate of subduction before interaction with the transition zone (compare dislocation-diffusion models with viscosity caps of ~$10^{24}$ and $10^{26}$ in Fig. 4). Subduction velocity is controlled by bending resistance at the trench, which is controlled by plate thickness and viscosity–here determined by plastic failure, rather than the viscosity cap. Therefore, early in subduction, the effective thickness of a slab has a larger impact on subduction velocity than maximum viscosity. This is illustrated by the fact that the diffusion-only slab subducts more slowly than thinner dislocation-controlled slabs (Figs. 3 and 4), despite the same maximum viscosity. These results imply that models

with high maxmimum viscosity/stress caps are best-suited to modeling the early stages of subduction, before the slab interacts with the transition zone.

## 5.2 Implications for the interpretation of analogue models

It can be very challenging to implement complex, non-linear rheologies in analogue modeling experiments due to the need for extensive scaling of material properties. Several analogue models have successfully incorporated temperature-dependent viscosity and a thermal gradient (Chemenda et al., 2000; Boutelier and Chemenda, 2003; Boutelier and Oncken, 2011), but the scaling of rheological properties in these experiments is less precise than what can be achieved through numerical modeling. In particular, as noted by Schellart and Strak (2016), in the models of Boutelier and Oncken (2011) and Boutelier and Chemenda (2003), the strength contrast between the lithosphere (a hydrocarbon mixture) and the asthenosphere (liquid water) is several orders of magnitude too high. Faccenna et al. (1999), Funiciello et al. (2008), Husson et al. (2012), and Chen et al. (2015), among others, use viscous materials like silicone putty to achieve a more realistic viscosity contrast between the lithosphere and sub-lithospheric mantle, but neglect thermal effects. These models sometimes incorporate a layered structure to capture brittle behavior in the upper part of the plate, but typically use a constant-viscosity material for the viscous portion of the lithosphere.

Our numerical models illustrate the extent to which slabs with temperature- and pressure-dependent rheologies can be approximated by constant-viscosity analogue models. Before the slabs in our experiments reach the lower model boundary, subduction velocity and the rate of internal dissipation of energy follow similar patterns over time in all models —speeding up at an increasing rate before abruptly slowing down as the tip nears the mantle transition zone —regardless of the complexity of slab rheology (Fig. 6). Although the slabs with diffusion and dislocation creep curl under rather than sliding forward, when bending resistance is reduced by a lower viscosity cap, the slabs slide forward like our reference model. This indicates that the qualitative behavior described by Funiciello et al. (2008) is not affected by increasing rheological complexity in the slab.

However, the entire length of the slab remains relatively stiff and negatively bouyant in analogue models, whereas realistic temperature-dependent implementations cause the slab to shorten as it is warmed by the surrounding asthenosphere. As discussed in Garel et al. (2014), the shortening of creep-governed slabs over time complicates the feedback between slab length and subduction velocity. The length of a slow-moving slab grows more slowly than that of a fast-moving slab, not only because it subducts more slowly, but also because a greater proportion of the slow slab assimilates into the mantle.

Analogue models typically also do not capture the contrast in bending resistance observed between the near-surface and deeper parts of slabs presented in this study. Subducted portions of our creep-governed slabs have bending resistances several orders of magnitude higher than the shallow areas of the plate near the trench, which are weakened by brittle failure (Table 3). This is even true of the model whose strength at depth is limited by a lower (~$10^{24}$) viscosity cap. This contrast in bending resistance causes the plates in our models to behave coherently and rigidly at depth but still bend readily at the trench.

## 6 Conclusions

The rheological laws implemented in subducting slabs in this study produce a range of slab viscosity structures, which in turn affect subduction dynamics. Resistance to slab bending plays a critical role in subduction dynamics at all stages. Initially, as the slab tip sinks freely, higher resistance to bending at the trench in the diffusion-only and reference models increases energy dissipation rate in the lithosphere (Fig. 7) and slows subduction (Fig. 4) relative to the models implementing dislocation creep. Bending resistance at depth also controls whether the slab flattens out forward or curls backwards after hitting the bottom of the model. This difference in slab geometry in turn results in dramatic differences in subduction velocity, slab dip, and trench rollback rate between the constant-viscosity slab and the stiffer, creep-governed slabs, in part due to the limited domain of our models. The slab controlled by diffusion and dislocation creep has a higher resistance to bending at depth, but a lower resistance to bending at the trench than the constant-viscosity reference model, due to the interaction between effective slab thickness and plastic weakening in shallow regions of the model. This results in higher subduction velocity in the creep-governed models despite much higher viscosity in the cores of these slabs than in the reference model.

The implementation of more complex flow-controlled rheologies also impact the feedback between subduction velocity, slab pull, and resisting forces relative to models with constant-viscosity lithosphere by shortening the effective length of the subducted slab. Most analogue models of subduction resemble the constant-viscosity slab modeled here in that they do not implement temperature- and pressure-dependant viscosity in the subducting lithosphere. Our results show that these models are likely to capture the qualitative behavior of slabs with a more complex rheology, but will not capture the feedback between slab length and subduction velocity, or the increase in subduction velocity due to reduction of bending resistance at the trench by plastic failure.

Models implementing only diffusion creep or dislocation creep, with brittle failure, predict unrealistically high viscosity in the core of subducting lithosphere. Plasticity helps to weaken slabs at the surface, allowing them to bend. However, high pressure prevents plasticity from lowering the effective viscosity of the lithosphere once it has subducted, resulting in very high slab stiffness. The dislocation creep mechanism is weaker than the diffusion creep mechanism throughout the slab over the duration of our experiments, causing the dislocation-diffusion slab to behave nearly identically to the dislocation-only slab. However, in the absence of a stress-limiting mechanism like Peierls creep, neither creep law implemented here is weak enough for slabs to deflect forward at the mantle transition zone, even with grain size reduction by several orders of magnitude. In our creep-governed models, the viscosity maximum is reached in nearly half the thickness of the lithosphere. Therefore, it is important to consider the realistic implementation of weakening mechanisms such as Peierls creep when designing numerical models of subduction.

## 7 Competing Interests

Susanne Buiter is an executive editor for Solid Earth. An independent editor supervised the peer-review process, and the authors declare no other competing interests.

## 8 Acknowledgements

The numerical experiments were performed with SULEC, a finite element software jointly developed by Susanne Buiter and

525 Susan Ellis. This research was funded in part by the German-American Fulbright Commission. We thank Fabio Capitanio and

two anonymous reviewers for their thoughtful and constructive feedback.

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

## Appendix A: Bending Resistance Calculation

In this section, we outline the calculation for resistance to bending. As discussed in the section 4.1, the rates of bending and stretching in a thin viscous sheet relate to the resistance to bending and stretching according to the following equation from Ribe (2001):

$$
\begin{bmatrix} N \\ M \end{bmatrix} = \begin{bmatrix} 4\mu H & 5\mu H^3 K/6 \\ \mu H^3 K/3 & \mu H^{3]}/3 \end{bmatrix} \begin{bmatrix} \Delta \\ \Omega \end{bmatrix},
\tag{A1}
$$

where $\Omega$ is the bending rate ($\mathrm{s^{-1}m^{-1}}$), $\Delta$ is the stretching rate ($\mathrm{s^{-1}}$), H is the thickness of the slab (m), $N$ is the stretching moment (Pam), and M is the bending moment ($\mathrm{Pam^{-2}}$), K is the curvature ($\mathrm{m^{-1}}$), and $\mu$ is the viscosity (Pas).

We rearrange the lower row of this matrix equation to solve for the bending resistance, $D$, which appears in the equation above as $H^3/3$:

$$
D = M/(\Delta K + \Omega).
\tag{A2}
$$

We solve for $M$, $\Omega$, and $\Delta$ at a particular snapshot in each model, once the model has reached its final geometry. Figure **??** shows the slabs at these time steps, colored by differential stress.

On each slab-perpendicular profile along the length of the subducting plate (an example is illustrated with a red line in Fig. A2 A), we calculate the stresses parallel to the slab, ($\sigma_{ss}$), velocities parallel ($\boldsymbol{u}$) and perpendicular ($\boldsymbol{w}$) to the slab (Fig. A2A), and dip. We then use these quantities to solve for bending and stretching rate, bending moment, and bending resistance.

On a particular profile, the bending moment is the integral of the slab-parallel component of differential stress multiplied by the distance from the zero-stress center line:

$$M = \int_{-H/2}^{H/2} z\sigma_{ss}\, dz$$


.        We sample stress along a profile perpendicular to the slab and find the stress minimum at the center of the profile. We then sum the slab-parallel component of stress times its distance from the zero-stress center line at each sampled point along the profile, multiplied by the spacing of the sampled points (dz = 1 km). The depth of the zero-stress center line varies along the length of the slab. The line used in our calculations is colored in white on Figure A2 C.

The bending rate of the slab at a particular point along the slab's length depends on the slab curvature and the slab-parallel velocity in addition to the second derivative of slab-perpendicular velocity with respect to distance along the slab, which captures the rate of change of slab curvature:

$$\Omega = -d\omega/ds, \tag{A4}$$

where,

$$\omega = dw_0/ds + Ku_0, \tag{A5}$$

where $u_0$ is the slab-parallel velocity at the center line. Approximating $u_0$ as constant, Equation A4 becomes:

$$\Omega = -d^2w_0/ds^2 + u_0 * dK/ds. \tag{A6}$$

We define positive velocity into the curve, so that when slab-perpendicular velocity has a positive second derivative, it contributes positively to bending rate.

There is significant uncertainty in the bending rate because the discrete sampling of slab location and velocity from our finite element models introduces small, semi-regular scatter into the values used (Fig. A2 D and E). These small variations become large variations when we take first and second derivatives of velocity and slab dip with respect to distance along the slab. We therefore fit curves to the sampled values (blue curve in Figure A2 D and E) so that the derivatives are more representative of the overall trends in velocity and slab dip. We fit polynomials to the measurements, and determine the degree of each
polynomial based on which fits match best visually and avoid unnecessary oscillations. We aim to match the curvature of the slab-perpendicular velocity (A2D) and the magnitude of the slab-parallel velocity in the vicinity of the stiffness measurement. Overfitting here is not an issue because we are not making statistical claims about these data; we are merely trying to achieve the best overall description of the pattern in the measurements for further analysis.

The uncertainty in the bending rate is greatest in areas where the bending rate is low. This makes the calculation of bending
rate, and by extension bending resistance, less precise in the deeper sections of models with overturned geometries, because

the bending is very slow in these areas. The precise fit to the slab-perpendicular velocities and the curvature of the slab can change the values by orders of magnitude, making the values of bending resistance reported here for deep, overturned slabs tentative approximations.

Despite these complications, our calculations match the order of magnitude expected for the stiffness of the reference slab, which we can calculate with the equation: $D = \mu H^3/3$ because the slab has a constant viscosity. The expected stiffness is $1.2 * 10^{37}$ $\text{Pasm}^3$ and we calculate a value of $6 * 10^{36}$ $\text{Pasm}^3$. The values in Table 3 are slightly lower across the board than the expected values for bending resistance at depth.

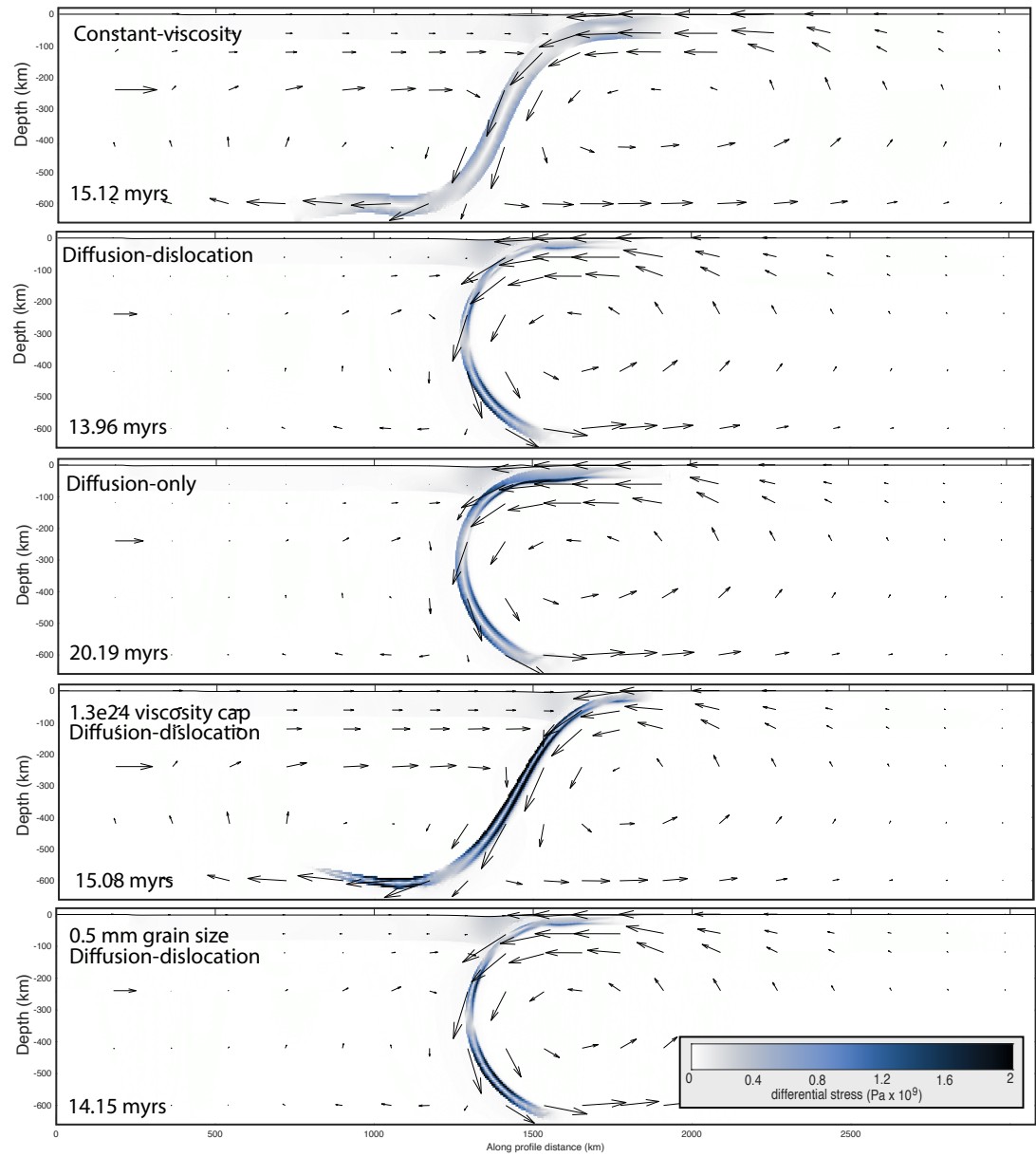

**Figure A1.** Images of the models for which we calculated bending resistance, colored by differential stress. Times on the lower left corner of each plot indicate at what elapsed time the measurements were taken.

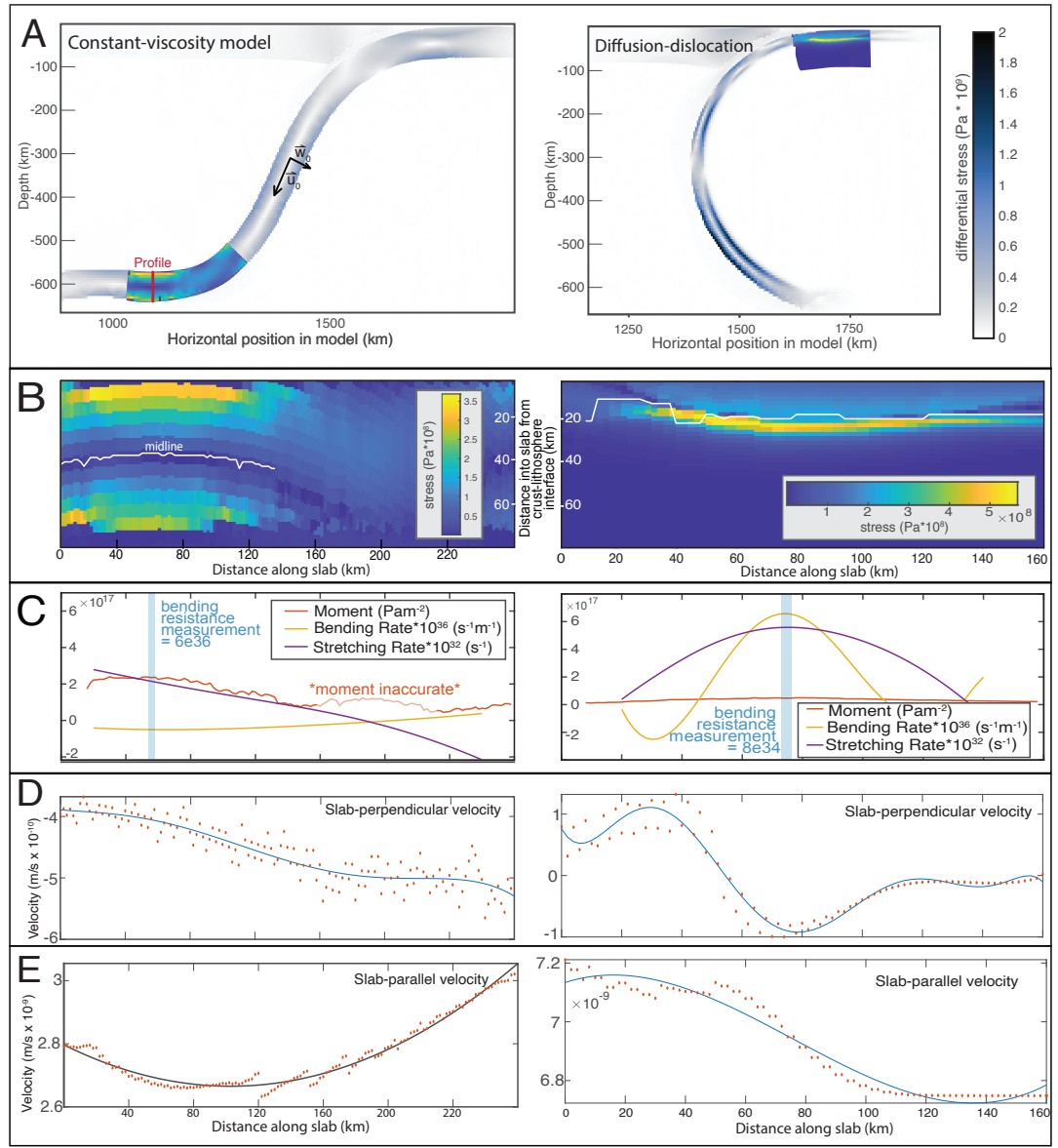

**Figure A2.** Panel A shows images of the constant $10^{23}$ Pas reference slab (left) and the diffusion-dislocation slab with a viscosity cap of $10^{26}$ Pas (right). Plots are colored by differential stress. Highlighted sections show where curvature, bending rate, etc. are calculated. These sections are colored according to the scheme in Panel B. For the reference slab, we highlight the calculation at the bottom of the model where the slab unbends, and for the diffusion-dislocation slab, we highlight the calculation at the trench, but calculations at both locations were performed for all models. B shows the analyzed sections of each slab, colored by slab-parallel stress, $\sigma_{ss}$, which is used in the calculation of bending moment. The approximate stress minimum at the center of the plate is traced by a white line. C shows bending rate, stretching rate, and moment, scaled by orders of magnitude to be visible on the same axes. The vertical blue lines indicate the profiles from which bending resistance was calculated, where bending rate is maximized. D and E Show slab perpendicular velocity and slab-parallel velocity, respectively, sampled from the model in red points, and the curves used to fit those points in blue.