# Peer review of "The influence of viscous slab rheology on numerical models of subduction"

_EGUsphere, 2023_

## Author Response (AR1)

We would like to thank the reviewers for extremely helpful, constructive feedback on our manuscript. We have done our best to implement suggested changes and clarify the goals of the study.

-Natalie Hummel, on behalf of the authors

**Responses to Comments from Reviewer 1**

*First off, we know fairly well that slabs appear overall more or less weak (say, 500....750 times the background mantle viscosity), but we do not know well why that is. There are a number of studies providing a range of constraints on the effective slab strength which seem not well represented, from the regional trench admittance work by Billen, to the global geoid modeling work by Zhong. The reviews of Billen (2008) and Becker and Faccenna (2009) give some references, for example, leading up to the modeling work of Cizcova who showed, in several studies, that some sort of plastic yielding is indeed required to weaken deep slabs from the high viscosities laboratory creep laws would imply.*

Zhong et al. (2008) and Mao and Zhong (2021) provide valuable evidence for a moderate ($10^2$-$10^4$) strength contrast between slabs and the surrounding mantle. We thank the reviewer for these suggestions, and have included citations on lines 387 and 388, where we discuss realistic values for the slab-mantle viscosity contrast.  We reference the work of Cizkova et al. (2002) on the importance of a stress-limiting mechanism (e.g. Peierls creep) (lines 82, 423, and 438 in the updated manuscript), but the reviewer is correct in pointing out that several other papers are also relevant to our discussion of slab weakening. We have included a sentence on the findings of Androvičová et al. 2013 on the importance of the viscosity of the upper slab interface on line 90.

*Now, as to the implementation of "realistic" laboratory-derived creep laws and plasticity, the key study by Garel et al. (10.1002/2014GC005257, 2014) needs to be discussed. This work really nicely explored a then-up-to-date view of what different deformation mechanisms do for certain assumptions, which is highly relevant here, including in terms of the updated phase diagram of Garel which builds on earlier such work by Stegman and Funciello and others.*

We agree with the reviewer that the findings of Garel et al. 2014 are highly relevant to our work. We have expanded both the introduction (section 1) and the discussion (section 5.1) to contextualize our results. Our stiff creep-governed slabs and 80 km thick overriding plate fall into their 'bent then inclined retreat' category. This gives us some insight into how our slabs may have behaved if the bottom boundaries of our models were not impenetrable. We address this in the second paragraph of section 5.2. Garel et al. also run models implementing Peierls creep, finding that it is active in large portions of the cold slab core, which is consistent with the high stresses reached in these areas in our models.

In the Introduction, we address their results on the effects of slab stiffness on slab-transition zone interactions (line 85), and the feedback between slab length and subduction velocity (line 89), as well as their results on the importance of Peierls creep (lines 94,95).

*What is also important is to realize how sensitive the balance of diffusion to dislocation creep is to partially quite uncertain creep law parameters, even if we assume that grain size is constant. I.e. there is no "right" upper mantle rheology for olivine, for diff or disl creep, nor for Peierls. The authors would obtain very different answers for wet and dry creep laws from Kohlstedt, for example, as is partially explored in Billen and Hirth 10.1029/2005GL023457 and 10.1029/2007GC001597), to note a few. Such uncertainties (in particular, activation volume, e.g. 10.1016/j.jog.2016.03.005) have led folks to often parameterize dislocation/diffusion creep with a simple transition stress or strain (e.g. 10.1002/2013JB010408). This allows dialing in the non-Newtonian response and exploring the effects.*

We definitely agree that there will not be one 'right' upper mantle rheology. Accordingly, our goal was not to find a single most realistic rheology that is tailored to the mantle-lithosphere system, but to explore the implications of some of the most commonly used approaches to implementing slab rheology in numerical (and analogue) modeling studies. It's a valid point that the parameters used in this study are not the only reasonable choices for a subducting plate, but the parameters from Hirth and Kohlstedt (2003) that we have picked for this study are very commonly used by a range of numerical modeling groups. These particular parameters produce high viscosities within the slab that are the subject of this study and, to some degree, illustrate the relative importance of different creep mechanisms.

*Importantly, the diffusion/dislocation creep rheologies do not just modify the viscosity within the slab, but also in the surrounding mantle, which can strongly affect dynamic processes such as slab rollback (e.g. 10.1093/gji/ggw392). That is to say, we cannot always easily capture rheology by changing effective slab rheologies - this approximation can sometimes be OK, but can miss important parts of the partitioning of viscous dissipation in others. Figuring out if such mantle-based non-Newtonian effects matter for what sorts of grain sizes and Arrhenius laws assumed for the lab-derived creep laws has to be an important step for a study with goals such as yours.*

We intentionally implement a constant-viscosity mantle to avoid the complexities of mantle rheology that the reviewer mentions here and isolate the impact of slab rheology. It is true that altering the mantle rheology would likely have a strong effect on the models, especially given the large rate of energy dissipation in the asthenosphere in all models, but exploring these effects is a study in itself beyond the scope of this manuscript. A constant viscosity mantle allows us to include the energy dissipation in the mantle as well as provide thermal and isostatic boundary conditions to the slab, but restrict rheological feedbacks and thus isolate the effects of slab rheology.

*Second, the effects of bulk strength variations and of layered viscosities for the bending and rollback behavior have been explored extensively. I am not sure what the current study adds, honestly. There is some argument that the weak slab tip due to lab-derived creep laws matters for the dynamics, but I would question if we are not simply seeing the effects of the initial conditions plus some broad, bending-related rheology effects. This has been explored extensively in Di Giuseppe et al. (10.1029/2007GC001776) which, like the Garel study, seems like a major oversight to not discuss/build upon. As a minor point, the role of side boundaries for slab bending/folding, to which you allude briefly, was explored by Enns et al. (GJI, 160, 761) as also summarized by Billen (2008).*

The reviewer is correct in pointing out that our results need better contextualization by discussing the suggested studies. We have expanded on the introduction as well as on the discussion to achieve this. We now cite Garel et al. 2014, Di Giuseppe et al., and Androvičová et al, 2013, as well as Moa and Zhong, 2021 and Zhong et al, 2008 on realistic slab-mantle viscosity ratios.

Nevertheless, we believe that our study differs from previous work in several key respects. We use a simplified setup to isolate the effects of slab rheology on subduction dynamics, exploring the impacts of viscosity structure on bending resistance and mantle drag. Our energy dissipation analysis allows us to identify when subduction velocity is controlled primarily by drag in the mantle and when it is controlled by resistance to slab bending/stretching. We compare the importance of each of these resistive forces in models of varying rheological complexity. We also believe that the unrealistically high slab stiffnesses reached in numerical models implementing diffusion and dislocation flow laws are worth further discussion, given that this is still a common approach in numerical models of subduction.

Our analysis explores why constant-viscosity slabs behave differently from creep-governed slabs (which are more complex than layered slabs since they undergo shallow brittle failure and are temperature-dependent, producing a feedback between effective slab length and subduction velocity). These findings have implications for simple numerical models and for analogue models lacking layering or temperature dependence. We also discuss how diffusion and dislocation-controlled slabs likely differ from slabs on Earth, which informs the interpretation of many numerical models.

*I was entirely confused by the slab pull/viscous dissipation discussion and did not learn anything from it (might be my ignorance, sorry if I missed something). If the authors focus on such analysis, why not then redo the Conrad and Hager force balance analysis that was mentioned? I realize that geometries are evolving here, but if this is relevant, then there are a number of other semi-analytical studies that went beyond which can be considered, e.g. Ribe, Neil M. "Bending mechanics and mode selection in free subduction: A thin-sheet analysis." Geophysical Journal International 180, no. 2 (2010): 559-576 and Gerardi, G. and Ribe, N.M., 2018. Boundary Element Modeling of Two‐Plate Interaction at Subduction*

*Zones: Scaling Laws and Application to the Aleutian Subduction Zone. Journal of Geophysical Research: Solid Earth, 123(6), pp.5227-5248.,), and also numerous numerical work, e.g. Capitanio, F. A., Morra, G., & Goes, S. (2009). Dynamics of plate bending at the trench and slab‐plate coupling. Geochemistry, Geophysics, Geosystems, 10(4). For pure visco-plastic bending, also see 10.1029/2012JB009205.*

The primary goal of the dissipation analysis is to demonstrate which areas (slab, asthenosphere, crust) are responsible for the difference in behavior between the reference and creep-governed models. For instance, plots of dissipation rate vs. velocity show that the initially higher acceleration of the reference slab is due, at least in part, to higher drag in the asthenosphere, which we believe is caused by the longer slab tip. High lithospheric dissipation rates starting around 10 myrs also confirm that creep-governed slabs slow upon contact with the bottom of the model (representing a much-simplified 660 km discontinuity) because they are forced to bend (and are too deep to fail brittley, rendering them quite stiff).

We agree that our first submission would benefit from more comparison to existing analytical approximations. We have quantified the bending resistance of the creep-governed slabs more precisely where slabs bend at the trench and at depth in the mantle– using bending rate and bending moment – in order to compare more effectively to the reference model.

We have reworked the Force Balance and Energy Dissipation section so that it is organized by equations for slab pull, bending resistance, and energy dissipation rate, then comparison of the models at each stage of subduction, rather than by dissipation in each material (crust, asthenosphere, lithosphere). We believe this will make the goal of the analysis more clear.

*So, I do not think we learned much new when it comes to understanding the behavior of homogeneous or layered slab with different viscosity or rheology, e.g. compared to the Di Giuseppe paper, or the Capitanio work. If this is your focus, then I think you need to show how your results go beyond the phase diagrams documented for strong and weak slabs, including by Garel. If your focus is to explore how different laboratory-derived creep laws (and assumptions on grain size etc.) compound to give different effective slab rheologies (including different plastic-type rheologies), then you need to show the range of predictions and explore uncertainties and robustness there.*

*If you can address those two major issues, then I think this could be a nice contribution, and this should not be too difficult given the nice numerical setup you have. Again, I hope this is helpful.*

We thank the reviewer for their input, which helped us to sharpen our main findings and improve our comparison to previous studies. As we have outlined in our responses, we have reworked our introduction and discussion sections to better place our approach and findings in the context of previous studies. Our contribution is a simplified numerical setup that allows isolating the impact of much-used laboratory flow laws on numerical slab

dynamics. In our analysis, we connect two often-used approaches for analyzing subduction models: force balance and rate of dissipation of energy. Forces can result in different slab shapes and different internal deformation of the slab, depending on the rheology. By examining the rate of dissipation of energy, we can investigate the process of deformation within the slab (and the mantle).

Our primary contribution is the analysis of mechanisms by which diffusion and dislocation-controlled rheology changes subduction dynamics relative to constant-viscosity approximations. We find that subduction velocity is increased in creep-governed models by the reduced bending resistance at the trench due to plastic failure and lower effective slab thickness, whereas the bending resistance of the creep-governed models at depth is very high. We also find that the effective shortening of the hot slab tip in creep-governed models may increase subduction velocity. In contrast to previous studies (Garel et al. 2014, Di Giuseppe et al. 2008), we focus on implications for building realistic numerical and analogue models, rather than trench motion, the effects of plate age, or interactions with the transition zone.

In order to emphasize the effects of more complex plate rheology beyond the increase in slab stiffness, we havel run (1) two creep-governed models with grain sizes reduced (moderately, to 0.5 mm, and extremely to 0.005 mm) in order to reduce bending resistance, and (2) a model with the viscosity cap reduced such that the bending resistance is comparable to that of the constant 1e23 viscosity slab. The former models illustrate the limitations of grain size weakening in models with very high viscosity caps. The latter model, in comparison to the reference model, illustrates the impact of creep-governed slab structure independent of changes in bending resistance. The latter model reaches a much higher subduction velocity early in the experiment than the reference model due to more dramatically lowered bending resistance at the trench from plastic failure and lower drag on the slab tip.

We thank the reviewer again for their help in honing the focus of our manuscript and for directing us to valuable resources.

**Responses to Comments from Reviewer 2**

*In this manuscript, a series of numerical models have been conducted to study the free subduction dynamics with variable rheological flow laws for the slab, i.e. constant viscosity, a diffusion creep law, a dislocation creep law, or both diffusion- dislocation in parallel. This an interesting topic and provide some basic constraints for the future numerical study of subduction dynamics. I feel that the current version of the paper lacks some conclusive remarks, since the goal of this study, which is very good, is: "We hope that these experiments raise awareness of the limitations of using extrapolated flow laws in numerical models of subduction and initiate a discussion on high viscosity values reached in many models."*

*My major concern lies in the comparison of models with very different "effective" viscosity, or stiffness, of the slab. For example, the three models in Figures 2-3, using either constant viscosity or creep flow laws with the resulting viscosity contrast of several orders. In this case, it is not surprising to produce different subduction styles. I am wondering, the more interesting result may be the comparison of models with different rheological laws (constant viscosity, diffusion, or dislocation), but have similar effective viscosity or slab stiffness. In this case it may clearly demonstrate the difference among using these contrasting rheological models.*

*The authors have tried to make a more direct comparison among models with different rheological models but similar slab stiffness, as in the paragraph around Line 280. However, this test focuses on an end-member with rather high and naturally unrealistic slab stiffness; consequently, the constant-viscosity model failed with subduction stalled.*

*A major goal of the study was to demonstrate the extent of stiffness variation – and the resulting effects on subduction dynamics – between the reference slab and the slabs governed by standard diffusion and dislocation creep laws. However, the reviewer makes a very good point that the effect of slab stiffness on subduction dynamics is well-studied, and that it would also be interesting to examine the effects of more complex rheology, controlling for slab stiffness. As they mention, we did run a model with a comparable slab stiffness to the creep-governed slabs, in which subduction stalled. We argue that this illustrates how a creep-governed viscosity structure speeds subduction relative to a constant-viscosity plate of a similar viscosity and thickness. This is likely primarily due, as this reviewer points out, to the more pronounced thinning of the stiff core at the trench in creep-governed slabs.*

At the reviewer's suggestion, we have run another set of models with reduced stiffness to compare to our reference model. We include two models following diffusion and dislocation flow laws with reduced grain size to lower the viscosity of the slab. We discuss a slab with grain size reduced by an order of magnitude (5 mm to 0.5 mm), which has very similar properties and behavior to the original creep-governed slabs. The second model had a grain size of 0.005 mm. This is not a realistic grain size, so we did not include a detailed discussion or images in the main text, but an animation showing the morphology and viscosity structure is included in the supplementary materials. Despite the extremely fine grain size, this slab does not feed forward at the transition zone. This is discussed in the paragraph near line 430. We also discuss a model with a viscosity cap near 10^24, and an overall stiffness similar to that of the reference model. This slab behaves like the reference model at depth but like the creep-governed models at the trench because its effective thickness is less than 80 km. We discuss this model in the last paragraph of section 4.2.1 and the fifth paragraph of section 4.2.2.

*I understand that the viscosity calculated with general parameters of olivine flow laws (Hirth and Kohlstedt, 2003 or Karato and Wu, 1995) will be very high for the top lithosphere; but it*

*is dependent on the grain size of diffusion creep that may vary for orders. In this sense, I am wondering whether the final viscosity can also be decreased a bit. If the effective viscosity calculated with creep flow laws is decreased, and/or the constant slab viscosity is increased (the viscosity of asthenosphere may be also increased accordingly in order to keep a reasonable slab/mantle viscosity ratio; it is a constant value of 10^20 Pa s in this study). Then, it may be possible to obtain a similar stiffness of subducting slab with different rheological models. How do you think about it?*

There are many parameters with reasonably high uncertainties in the flow laws, and undertaking an exploration of all of these properties is beyond the scope of the study. However, varying just grain size is a nice suggestion, as this property is naturally variable and has been proposed as an important weakening mechanism (Karato et al. 2001). We have run two additional experiments with reduced grain size and bending resistance. Due to the high (10^26) viscosity cap in our models, however, neither slab bent forward like the reference slab at the transition zone, despite the extremely low grain size (5 microns) implemented throughout one of the modeled slabs (see lines 430-440). This supports the conclusions of Cizkova et al, 2002, that grain size weakening plays a smaller role in the bending resistance of a slab at depth than the stress cap used in the Peierls creep law.

We also ran a model with a reduced viscosity cap to explore the behavior of a slab with generally similar structure to the creep-governed models but similar stiffness to the reference model. This model is discussed in the paragraph near line 330 and the paragraph near line 358, as well as in the Discussion.

*Another point is about the "Comparison to realistic slab rheology". As this study and many previous studies (not list here) have shown, the slab/mantle viscosity ratio should not be greater than 1000 (a summary could be found in Li and Ribe, 2012, JGR) in order to be consistent with the natural slab morphologies, because the geometry with slab bending backward is rather rare in nature (the Indian/Neo-Tethys case may be the only one; Van der Voo, 1999). However, the viscosity calculated with experiment-based creep flow laws is generally very high. Thus, the weakening mechanism should be important, as Gerya et al. (2021, Nature) has shown. I think the effective viscosity of the outer-rise bending zone, rather than that of the horizontal plate or subducted slab, controls the force balance of subduction. In this study, the subduction velocity in the models with creep flow laws, especially that with dislocation, is higher than the model with constant viscosity, although the latter has low slab stiffness. The authors have attributed this phenomenon to the high resistance due to the high viscosity slab tip in the constant viscosity model. I agree with this point; but I would also consider another point about the plate bending resistance. Although the plate with creep flow law produce higher viscosity generally, the viscosity at the bending zone could be low, as shown in the Figure 5B. Thus, if comparing the effective viscosity at the bending zone in different models (Figure 3), the bending resistance in the creep models may be lower (I am not sure; requires a calculation).*

This is a great point, and we have calculated bending resistance more precisely than we did in the original manuscript by calculating bending moment based on stresses in the slab and calculating curvature rate based on the spatial curvature and the subduction velocity and trench rollback rate. The calculation is subject to considerable uncertainty due to the limited resolution of our models, but still allows us to generally compare the stiffness of the slab at the trench with the stiffness of the slab elsewhere and with the reference model. Values are shown in Table 3. The detailed description of the calculations is given in Appendix A and the results are discussed in section 4.

The reviewer is correct that the reduction of slab stiffness at the trench increases subduction velocity of the creep-governed slabs relative to the reference model. We believe that the role of the slab tip in subduction dynamics when a constant viscosity slab is used (i.e. in analogue modeling experiments) is generally overlooked and is worthy of exploration. We base this assertion on our observation that, at a constant subduction velocity, dissipation rate is higher in the asthenosphere in the reference model than it is in the creep-governed models, suggesting higher drag in the reference model. However, the reviewer is correct that, according to the analysis of Ribe (2010), drag likely has a small effect relative to the bending resistance. We explore this in paragraph 4 of section 4.2.1.

*As has been argued in this study, the weakening of slab plays a critical role in the subduction models. I am wondering whether you can give more discussion of this point, which should contribute a lot for the understanding and future study.*

Based on this reviewer's helpful comments, our revised discussion emphasizes how reduction of effective slab thickness/bending resistance at the trench plays an important role in weakening subducting lithosphere. These points are illustrated by the comparison in behavior between the creep-governed slabs and the constant-viscosity slab. The constant-viscosity slab has a higher stiffness in most regions, but subducts more slowly than the creep-governed slabs because its bending resistance is reduced less by plastic failure at the trench. This is discussed in the last paragraph of section 4.2.1. The models also show that weakening mechanisms beyond those implemented here are required to achieve realistic slab behavior at depth. We have emphasized this point in section 5.1, where we discuss the studies of Karato et al. 2001, Gerya et al. 2021, Garel et al. 2014, and Cížkova et al, 2002, which illustrate the effects of localized grain size reduction and Peierls creep.

*By the way, Figure 5 is very informative; but it just provides the information for one model with diffusion and dislocation in parallel. I would suggest adding the information for the other two models in order for a direct comparison.*

We thank the reviewer for the good suggestion. We have added insets to this figure

showing the viscosity reduction at the trench in four models: the reference model, the diffusion-only model, the diffusion-dislocation model, and the model with a reduced viscosity cap. The strength vs. depth curves for the three primary creep-governed models were already shown along profile C, and the diffusion creep strength curve in plots A and B provide insight into the strength of a diffusion-only slab in those areas (although they were calculated using strain rates and temperatures from the diffusion-dislocation model).

*Some specific points:*

*L9, "(5mm) grain size" Is this value used in all the numerical models? I remember comparing the grain sizes in Karato and Wu (1995) and Hirth and Kohlstedt (2003). They may require different grain sizes to obtain the similar viscosity.*

Yes, 5 mm is the grain size used in all of the models. But note that we have included new models with lower grain sizes (0.5 mm and 0.005 mm).

*L14, "Our models also demonstrate a feedback between effective slab length and subduction velocity." You may state clearly what kind of feedback.*

We have changed this text to: "Our models demonstrate that higher subduction velocity causes a longer effective slab length, increasing both slab pull and asthenospheric drag, which, in turn, affect subduction velocity. Numerical and analogue models implementing constant viscosity slabs lack this feedback, but…"

*Figure 1, "initial slab pull" lacking unit for the value.*

We have added "N" to the figure.

*Equation 1, "-E + PV" should be "-E - PV"?*

The reviewer is correct, we have fixed equation 1.

*L132, "All models simulate a linear viscous asthenosphere, with a viscosity of 10^20 Pas" I agree that a constant viscosity for the asthenosphere is important for the comparison among models. If using similar creep flow laws, will the resulting viscosity be higher than this value?*

Likely not much higher. The base of the lithosphere reaches the viscosity minimum of 10^20 Pas in all creep-governed models (Fig. 5), suggesting that the mantle viscosity from the creep laws might even have been lower than the constant value used, particularly in areas of high strain rate around the slabs. In the subduction models run by Garel et al. 2014, the viscosity of the surrounding mantle is determined by creep laws and does not get much higher than 10^20 Pas.

*L143, "a free-slip bottom boundary" Why not using a rigid boundary as more consistent with the laboratory experiments, e.g. Schellart, 2008, G3. Surely that the nature is more complex.*

Our bottom boundary is fixed in the vertical direction (no flow of material across this boundary), but material can freely move horizontally along the bottom boundary. This is a common approach in numerical models of subduction limited to the upper mantle (e.g. Di Giuseppe et al. 2008). It is an interesting question whether a horizontally fixed or free slip condition would best approach a natural setting within the limitation of a non-permeable boundary. Indeed, our setup resembles a laboratory setup in not allowing material to cross the bottom boundary. The material simulating the mantle in laboratory experiments may 'stick' to the bottom boundary or show some horizontal movement, depending on the experimental setup. For instance, see experiment 2–with material stationary on the bottom boundary–and experiment 4–with the plate sliding along the bottom boundary– in Funiciello et al. 2006 (doi: 10.1029/2005JB003792). In general, we think that experimenting with a different type of bottom boundary condition is beyond the scope of this manuscript.

*Figure 4, Why does the subduction velocity keep increasing in the creep models until the end of modeling? If the plate is long enough, will it just increase all the way?*

The plate velocity would likely not continue to increase indefinitely. The rate of energy dissipation in the lithosphere due to bending decreases dramatically once the plates have overturned because they do not need to unbend at the transition zone. This allows the slabs to accelerate considerably. Velocity will reach a steady state when the slab pull force–which becomes approximately constant once the plate reaches the bottom of the model–equals the resisting forces from deformation of the mantle and the plate interface, plus bending and stretching of the subducting plate. The resistance in the mantle and along the plate interface will continue to increase (as shown in Figure 7) with subduction velocity, whereas slab pull should not. Therefore, presumably, the plate would eventually find an equilibrium velocity, though our models do not tell us what that velocity would be. We have mentioned this paragraph 3 of section 4.2.2.

*L183, "The slab has a constant viscosity over this interval, but lower strain rates in the middle of the slab result in lower stresses and a dip in strength centered approximately 18 km into the slab." Do you mean the minimum value at about 21 km in Figure 5A, blue line? You may plot the strain rate diagram to see the obviously low strain rate layer.*

Yes, we are referring to the dip in strength around 21 km deep in Figure 5A. We had been referring to the depth of the strength dip in the lithosphere, but the depth axis on the plot is with respect to the top of the crust, so '18 km' was confusing, as the dip appeared to occur below the 20 km mark. We have adjusted the language in the text: "and a dip in strength centered approximately 20 km into the slab." We have also added a strain rate curve on plot A of Figure 5.

*L256, "Therefore, the lower maximum subduction velocity in the reference model is primarily due to greater resistance in the lithospheric mantle, relating to the length and shape of the slab." It seems to be different from the argument somewhere, see the above major concern.*

The creep-governed slabs have a higher resistance to bending at depth, but, since they overturn at the transition zone and are able to maintain a constant curvature throughout the upper mantle, the rate of energy dissipation in the lithosphere in these models is actually less than the rate of energy dissipation in the constant-viscosity slab, for a given subduction velocity.

We have reworked the discussion of energy dissipation and subduction dynamics extensively and hope the section is more clear in its current form.

*L246, "mantle lithosphere 6" what is the "6" for?*

The number 6 here was meant to be a reference to Figure 6, but we misformatted it in the Overleaf template. We have fixed this typo.

*L249, "The rate of energy dissipation due to bending is:" I think the following Equation 5 represents the bending force. Is it equivalent to the energy dissipation rate?*

Bending force is related to, but not equivalent to energy dissipation rate; energy dissipation rate also depends on strain rate. The equation was intended to be the equation for bending dissipation rate from Conrad and Hager, 1999. There was, however, a typo in the equation--the velocity should be squared. Furthermore, Capitanio et al. 2009 demonstrated that the velocity dependence in this equation is inaccurate, so we have removed this equation and instead present the equations relating bending moment and resistance to bending (which we calculate more precisely for the creep-governed slabs) and bending rate from Ribe (2001). See equations 5 and 6 in the revised manuscript, the extensively reworked energy dissipation section and the supplement containing the detailed method for the new calculations.

*L305, "These studies suggest that, for a sub-lithospheric mantle viscosity of 10^19-10^20 Pas, slab viscosity should not exceed 10^23 Pas" Yes, but the mean viscosity of sub-lithospheric mantle could be higher; the top layer may be weaker, the other part could be higher. I have no idea about this effect.*

The reviewer makes a good point that potentially variable viscosities in the sub-lithospheric mantle make it difficult to name a single slab-to-mantle viscosity ratio. Though this makes it difficult to compare viscosity contrasts in our model to Earth, it is slightly easier to compare our model to other models that have also assumed a constant viscosity mantle. Among the models discussed, many of which assume a constant viscosity mantle, our creep-governed models do not match those experiments which provided the best fits to real slab morphologies. We believe that adding a model with a more realistic, non-linearly viscous

sub-lithospheric mantle is beyond the scope of the manuscript, as we are focused on the effects of slab rheology in a simplified subduction setup.

*L323, "tnhan"?*

*We have fixed this typo.*

*Second 5.1, Very descriptive; but it is better to have some conclusive remarks; otherwise, the readers do not know what are the new points from this study.*

We have improved the emphasis on the most important findings based on the reviewer's suggestion. We have added three paragraphs at the end of section 5.1 emphasizing that, without weakening mechanisms beyond those implemented in this study, slabs governed by diffusion and dislocation creep laws with moderate grain sizes – as are often used in numerical models – appear unrealistically stiff, impacting subduction dynamics. Even models with unrealistically low grain sizes appear unrealistically stiff at depth due to the very high stress cap used in our models, illustrating the importance of Peierls creep. Additionally, temperature-dependent plates have a more dramatically reduced effective thickness at the trench than constant-viscosity slabs, and potentially less drag around the slab tip, increasing subduction velocity in creep-governed models.

**Responses to Comments from Reviewer 3**

*The paper 'The influence of viscous slab rheology on numerical models of subduction' by Hummel, Buiter and Erdős addresses the impact of the common choice of different constitutive laws on subduction modelling. The paper tests a range of rheologies from Newtonian, proxy for the laboratory models, to more complex, and more realistic, non-Newtonian rheologies and different plastic flow laws. The models capture a range of behaviour addressed in past efforts and shows that although the morphology and stokes sinking velocities may appear the same, the dissipation may be different. This has an impact on the inferences we draw on the stiffness of the natural slabs, when the morphology and subduction rates are the only observable fit, showing that these provide non-unique constraints.*

*The paper is well written, and I don't have major concerns, just some discussion points and some more specific comments, below. However, I would recommend the authors to stress what is the real novel conclusion of this work. Many aspects of this work have been addressed before, then, what we learn from this work could be better illustrated.*

*Melbourne*

*30 August 2023*

*Fabio A. Capitanio*

*Some discussion points:*

*As a general point, I would like to suggest being more specific with the terminology, as "weak" and "strong" have been used in the past with little clarity. Plates are rigid, but they are not… In fact, it depends on the moment applied and the plane of application. Plates resist in a way to stretching and in another to bending, different mechanical properties. Also, plates do not bend under the torque applied in the plane of the Earth's surface, as the major length controlling their resistance is their width, therefore they are rigid. However, when the bending moment is applied in the perpendicular plane, vertically, the thickness and bending resistance drops, then plates bend easily. This has not necessarily much to do with the viscosity. Nor has the strain rates. In fact, the bending profile is such that there is a neutral plane where the stress is minor, where high viscosity/low strain rates is maintained, whereas the stresses/strain rates maximise in the outer layers. Therefore, the strain rates on Earth are not necessarily indicative of the whole plate yielding. So, I have nothing against the plate viscosity used here, since the thickness of the high viscosity layered is smaller than other works, it seems (resolution?). Then, the first-order behaviour is captured by the trade-off between thickness and viscosity and the realistic rheology supports the validity of the results. Perhaps, better than discussing the viscosity of the lithosphere, it would be clearer to discuss their mechanical resistance to bending.*

We agree that terminology can be unclear. We used "stiffness" to refer to bending resistance, but "resistance to bending" is more precise, so we now use this term instead, and define the concept earlier in the text, on line 74, in the introduction. We have calculated the bending resistance on several profiles for each plate (Table 3) and discuss these values in section 4.2. We also define the dimensionless stiffness discussed in Ribe 2010, though we primarily refer to bending resistance (Pasm^3) throughout the manuscript.

*The discussion (section 5.1) could be improved describing what the dissipation means for slabs. High dissipation implies large strain. **The total dissipation is equal to the potential energy released, then if the dissipation was dominant in the lithosphere, that is, the bending, then the mantle would deform less, and slabs slow down, as the potential energy is dissipated into deforming the slab, not the mantle.** This is, in few words, the conclusion in Conrad and Hager, 2001, and also an end member case presented by Neil Ribe in his fantastic work, that is infinite viscosity contrast. In this case, the bending rate depends on the viscosity of the slab. Yet, the major point against this case, i.e., dominant lithospheric dissipation, is that the dissipation in the mantle is a requirement of the Stokes regime. We model subduction in this regime, implying that the dissipation in the lithosphere must be close to/negligible, therefore, plates must bend easily, stretch little and propagate stresses to the surface, acting together with mantle tractions.*

Indeed, we agree with this and we thank the reviewer for the additional points suggested for the discussion. We have included an explanation of the way in which increased partitioning of energy dissipation into the lithosphere slows slabs down in the paragraph near line 250 in

section 4.1. We have also covered the distribution of dissipation between the mantle and subducting lithosphere more thoroughly in section 4.2.2.

*Some comments on the text*

*Line 4 Simplified, rather than simple?*

We have changed this line based on the reviewer's comment.

*Line 6 "both in parallel" is misleading, if it is meant to be "both at the same time" then change.*

We have updated the abstract to read "both simultaneously". We have also changed "in parallel" to "simultaneously" on lines 52 and 205, and to "together" on line 101.

*Line 8 Sentence is not clear. Dominates over…?*

We have changed it to: "We find that dislocation creep is the primary deformation mechanism throughout the subducting lithosphere with moderate (5 mm) grain size in the upper mantle."

*Line 14-16. The conclusion is important for the inferences on slabs' viscosity on Earth, when this is based on the morphology only. I think this is worth mentioning*

We have changed the text here from "Numerical and analogue models implementing constant viscosity slabs lack this feedback, but still capture the qualitative patterns observed in more complex models," to: "Numerical and analogue models implementing constant viscosity slabs lack this feedback, but still capture the morphological patterns observed in more complex models."

*Line 19 the calculation of maxwell time is fine, but should be used to make a point, otherwise remove from the introduction. It's an introduction, after all.*

Based on the reviewer's suggestion, we have moved the discussion of Maxwell relaxation time to the paragraph starting on line 145 in Section 2, where we explain the material properties of the slab, in order to justify a visco-plastic approximation.

*Line 40. In fact, in Capitanio et al., 2008, we have shown that the constant viscosity slab reproduces only first-order behaviour, but does a bad job with stresses and strain. Then, we introduced a layered slab to capture the effect of bending stresses and weakening of the outer slab around a stress-free core, to simulate the effect of more complex rheology. This was the best way to prove the relevance of the mechanical properties, that is, the resistance to bending, as opposed to the viscosity only.*

We have explained the difference between slab viscosity and resistance to bending and stretching in section 4.1. We have discussed the layered viscosity models in Capitanio et al. 2009 in the Introduction:

"... Capitanio et al. (2009) demonstrate that plates require a strong, thin (less than the typical thickness of an oceanic plate) core to bend readily at the trench, yet maintain sufficient resistance to stretching to transmit stress from the slab to the surface."

*Line 60. This is a good point where the difference between viscosity and mechanical properties could be discussed. Perhaps move here and expand the lines 64-66, where this is touched upon. In this sense, the different viscosities can be reconciled.*

We have reworked several paragraphs in the Introduction to make the distinction between viscosity and mechanical properties more clear. The paragraph beginning at line 71 discusses the effects of bending resistance on slab behavior, and the following paragraph discusses previous work on the differences between constant-viscosity models and more complex slab structures, including the layered slabs from Capitanio et al. 2009 and the creep-governed models from Garel et al. 2014.

*Equation 2 is incorrect, the second term in the rhs is adimensional, please, amend.*

We would like to thank the reviewer for catching this mistake. Equation 2 had been: rho(T) = $\rho 0 + \alpha(T1-T0)$. We had omitted the $\rho 0$ in the final term:

We have changed the formula in the text to: $\rho(T) = \rho 0 + \rho 0*\alpha*(T1-T0)$.

*Line 250. I'd recommend readjusting this part. In fact, the scaling we provided in Capitanio et al., 2007 (note: not 2008) for the dissipation is not the one in eq. 5. The point we made in the 2007 paper is that the velocity-dependent formulation used here, after Conrad and Hager, 1999, is misleading and leads to higher dissipation for faster plate velocities. This is an artifact, as the increased plate velocity (or convergence velocity) does not reflect increased sinking velocity, that is, there is no increase in potential energy. Therefore, we proposed in the 2007 paper a parameterisation for the dissipation based on the slab mass, not the velocity. Additionally, we followed this up in Capitanio et al., 2009, where we show in fig. 9 that the models are best fit in this way, as opposed to the Conrad & Hager 1999's velocity-dependent parameterisation.*

We have removed Equation 5, instead discussing bending resistance in more detail in section 4.

*In fact, this seems to be the result here, as shown in figure 7a to c: because the system is driven by buoyancy only, the dissipation in the asthenosphere increases with increasing velocity, however this is not the case in the lithosphere (c), where the dissipation ends up being very low. At the steady state, roughly from Figure 7f onwards, the dissipation in the*

*lithosphere is very low, in all models shown, except constant viscosity. This is rather in agreement with our findings that the dissipation in the lithosphere is low <40% in the constant viscosity case, and drops to ~20% (not 80% as stated in this submitted manuscript) in the layered viscosity, which localizes stress in the core, strain in the outer layers. The high dissipation in the crust is a bit puzzling, though.*

In the text, when we cite 80% dissipation due to lithospheric bending, we mean that out of the total lithospheric dissipation in experiments by Capitanio et al. 2007, >80% was due to bending, and <20% due to stretching. It was not meant to be a comparison between dissipation in different materials. We have clarified this point in the text:

"In models presented by \citet{CapitanioEtAl2007}, bending plays a larger role in the dissipation of energy than stretching in stiff slabs, accounting for \textgreater 80\% of the total dissipation." (Line 265)

We have also emphasized that the rate of energy dissipation in the lithosphere is not a strong function of subduction velocity in paragraph 5 of section 4.1. We have mentioned (near line 353) that the low lithospheric energy dissipation rates in our creep-governed (layered) models are consistent with the findings of Capitanio et al. 2009. However, we note that the very low energy dissipation rate in the creep-governed slabs late in the experiments at steady state is likely mostly a result of the overturned geometries of these slabs, rather than a result of the structure of the slab itself, as shown in Capitanio et al. 2009.

*Section 5.1 This is the section where the outcomes could be discussed in terms of what they mean for real slabs. See major comments above.*

Though we are hesitant to draw strong conclusions about the rheological structures of real slabs from our simplified models, it does seem clear that Peierls creep, grain size reduction, or other unaccounted-for mechanisms weaken slabs considerably relative to the creep mechanisms implemented here. Furthermore, grain size reduction alone may not weaken the lithosphere sufficiently if viscosities are allowed to get very high within the slab. We emphasized this point in the seventh paragraph in section 5.1. In this section, we also discuss the role that plastic failure plays in allowing slabs to bend at the trench, as well as the increase in slab stiffness with depth as plastic failure becomes inactive.

*Also, there is no case of overturned slab on Earth. The Indian case looks like it, but the overturning is due, there, to the advancing trench. There are many reconstructions that can show the slab was extended to the transition zone (straight), then overturned when the trench started advancing. I have tried to make the same point years ago, that of an overturned slab, then the wise reviewer #2 told me off. I think s/he was right, though, but I'll let the authors discuss their point.*

We have qualified our point – that overturned slabs in our models subduct very quickly and the only slab on Earth that appears to be overturned also happens (perhaps by chance) to

have had a very high subduction velocity – by noting on line 402 that Qayyum et al. 2022 believe this slab overturned late in subduction due to a period of trench advance: "However, some authors (Qayyum et al., 2022) argue that this slab overturned recently due to a period of trench advance, limiting the analogy to our models."

*Line 360 as said before, "soft" is not a very clear definition*

We have reworked this section and removed this line, but elsewhere we have made sure to use the terminology "reached asthenospheric viscosity" (Figure 2 caption).

---

## Author Response (AR2)

Dear Copernicus editorial team,

We have addressed the concerns of Reviewer 2 regarding the impacts of our closed model design, and have made several small edits to the text to fix typos or improve clarity, including a small stylistic change to the abstract.

**Comment from reviewer 2: This is my second review of this paper and I think my previous concerns have been mostly clarified. I still have a concern about the largely curved slab (slab rolling back to the lower side) in many models with relatively high slab stiffness, because this case is rarely observed in nature. I am wondering whether the 'impermeable' lower boundary condition strongly controls this phenomenon. Such a style of slab geometry and kinematics is mostly predicted in the 3D analogue models with rigid lower boundary or 2D/3D numerical models with either free slip or rigid lower boundary. In contrast, most of the models with a phase transition at 660 km do not produce such a strong rolling back structure (just my impression; should be clarified). On the other hand, the interaction between subducting slab and 660D strongly controls the slab morphology as already discussed in several review papers. Thus, I would suggest discussing the limitations of lower boundary condition (as the free slip or rigid boundary is a great simplification of the 660D). As a summary, this study clarifies many detailed aspects of plate rheology on the subduction dynamics; thus, I think it should be worthy for publication.**

We agree that the free-slip boundary at 660 km affects model behavior and should be addressed more explicitly in the text. We have improved our discussion of this effect in two locations:

On lines 343 through 345, after discussion of the high subduction velocity reached by the overturned slabs, we have included the sentence:
*"The free-slip boundary at the bottom of the models likely exacerbates this acceleration, as other implementations of the transition zone might provide more resistance to lateral sliding of the slab at 660 km."*

And in the discussion, near line 407, we have included the paragraph:

*It should be noted here that our models approximate the base of the upper mantle as a hard boundary, which undoubtedly has an impact on slab morphology and subduction dynamics once the slab tip reaches the bottom of the model. For this reason, divergence from realistic behavior at the later stages of our experiments cannot be entirely attributed to high slab stiffness. If the models presented here had a viscosity contrast at 660 km depth, rather than a hard boundary, the creep-governed slabs may have penetrated the mantle transition zone. Sufficiently stiff slabs in the models of Garel et al. (2014) approach the transition zone bent, like our slabs, but, upon reaching 660 km depth, continue vertically downwards or undergo trench retreat to bend forward. On the other hand, in the 3-dimensional models of Stegman et al. (2010), which also simulate a viscosity contrast at 660 km depth, slabs with sufficiently low Stokes buoyancy, and with comparable stiffness to our creep-governed slabs ($\eta_{slab}/\eta_{mantle} (H_{slab}/H_{mantle})^3 = 1\text{-}100$), overturn, similar to our models. It is difficult to know which morphology our slabs would exhibit if our models had included a lower*

*mantle. Regardless, our overturned, creep-governed slabs appear unusually stiff, despite moderate (< 80 km) effective thicknesses.*

We again thank the reviewer for their insightful feedback. It has considerably improved the quality of our manuscript. We would also like to thank the editorial team for their help throughout the revision process, and Reviewer 1 for their feedback on our first revision.

-Natalie Hummel, on behalf of the authors